# SCISSOR: a framework for identifying structural changes in RNA transcripts

Hyo Young Choi[1,2,3], Heejoon Jo[1,2], Xiaobei Zhao[1,2], Katherine A. Hoadley [4,5], Scott Newman [3], Jeremiah Holt[1], Michele C. Hayward[5], Michael I. Love[4,6], J. S. Marron[7] & D. Neil Hayes [1,2,3✉]

High-throughput sequencing protocols such as RNA-seq have made it possible to interrogate the sequence, structure and abundance of RNA transcripts at higher resolution than previous microarray and other molecular techniques. While many computational tools have been proposed for identifying mRNA variation through differential splicing/alternative exon usage, challenges in its analysis remain. Here, we propose a framework for unbiased and robust discovery of aberrant RNA transcript structures using short read sequencing data based on shape changes in an RNA-seq coverage profile. Shape changes in selecting sample outliers in RNA-seq, SCISSOR, is a series of procedures for transforming and normalizing base-level RNA sequencing coverage data in a transcript independent manner, followed by a statistical framework for its analysis (https://github.com/hyochoi/SCISSOR). The resulting high dimensional object is amenable to unsupervised screening of structural alterations across RNA-seq cohorts with nearly no assumption on the mutational mechanisms underlying abnormalities. This enables SCISSOR to independently recapture known variants such as splice site mutations in tumor suppressor genes as well as novel variants that are previously unrecognized or difficult to identify by any existing methods including recurrent alternate transcription start sites and recurrent complex deletions in 3′ UTRs.

[1] Department of Medicine, University of Tennessee Health Science Center, Memphis, TN, USA. [2] UTHSC Center for Cancer Research, University of Tennessee Health Science Center, Memphis, TN, USA. [3] Department of Computational Biology, St. Jude Children's Research Hospital, Memphis, TN, USA. [4] Department of Genetics, University of North Carolina, Chapel Hill, NC, USA. [5] Lineberger Comprehensive Cancer Center, University of North Carolina, Chapel Hill, NC, USA. [6] Department of Biostatistics, University of North Carolina, Chapel Hill, NC, USA. [7] Department of Statistics and Operations Research, University of North Carolina, Chapel Hill, NC, USA. ✉email: neil.hayes@uthsc.edu

Many human genes differ in function through various expression changes in mRNA product[1–6]. For example, tumor-suppressor genes and oncogenes can lose or gain function through aberrant splicing, gene fusions, duplication, short insertions, and deletions (indels), or overexpression[7–10]. These alterations are emerging as relevant targets of therapy, and thus the systematic discovery of such alterations is critical[11,12]. Many computational methods can identify mRNA aberrations from RNA-seq, but most use a limited subset of the total RNA-seq data available. For example, isoform detection methods that emphasize known transcripts such as DEXSeq[13] and SpliceTrap[14] collapse reads aligning to exons to a single value without incorporation of spliced reads. Alternatively, transcript agnostic methods such as MISO[15], rMATS[16], and DiffSplice[17] generally ignore non-junction-spanning reads, and analysis is restricted to pre-specified events of interest such as intron retention or exon skipping[18]. In either case, the transformed RNA data object is a greatly compressed representation of the true underlying data, and this can obscure various types of RNA variation. Therefore, there is an unmet need for unbiased methods that work on a less-compressed—or base resolution—representation of the transcriptome and thus could have greater power to reveal novel biological insights[19,20].

Here we present SCISSOR, an approach for systematic discovery of changes in mRNA expression including alternative splicing, intron retention, de novo splice sites, intra-/intergenic deletions, and alternative transcription start/termination (ATS/ATT) sites. RNA changes may be somatic, as the result of driver or passenger mutations, germline variants, or non-genetic events resulting from epigenetic regulation of alternate isoforms. In contrast to existing methods, SCISSOR uses base-resolution read coverage data that provide a rich and comprehensive landscape beyond gene/exon expression values or splice junctions. SCISSOR aims to detect structural variation, or differential coverage patterns, across RNA-seq cohorts without any underlying assumption of the mechanism driving the coverage variation. This enables us to reduce our dependency upon known gene models and increase our potential to confidently identify otherwise obscured genetic aberrations. As a proof of principle, we used SCISSOR on a cohort of 522 TCGA head and neck squamous cell carcinomas (HNSC) to identify known abnormalities in cancer-related genes[21]. We then identified novel aberrations missed by previous studies on the same data set including alternate splice isoforms, fusions, and intragenic deletions. Finally, we applied SCISSOR genome-wide to identify a novel set of genes with strong evidence of aberrant structure in HNSC.

## Results

SCISSOR relies on the post-alignment pileup format to represent the base-level coverage of each individual locus of interest[22]. The method allows for transcript-independent analysis based on genome alignments, although for the sake of visualization, convenience, and interpretation in the current example we limit this analysis primarily to annotated regions of the genome and exclude most intronic bases (see "Methods" section). Base-level read-depth represents an easily interpreted view of coverage shape of genes, with the $y$ axis proportional to per-base expression and the $x$ axis showing the genomic position (see "Methods" section and Fig. 1a). Our underlying assumption is that the majority of samples from the same tissue type will show uniform coverage and abnormal expression patterns will present as diverse shape changes that could be the result of various mutational mechanisms such as exon skipping, intron retention, gene fusion, deletion, or internal tandem duplication (Fig. 1a). As a tool for the systematic discovery of a variety of genetic aberrations, we use SCISSOR to interrogate shape changes where the aligned coverage shape is significantly different from the majority of samples.

**Pre-processing procedure**. Before the main shape change detection procedure, normalization and variance stabilization are performed to generate more symmetrical distributions of the SCISSOR test statistic and to facilitate interpretation of results across genes. Because SCISSOR is unlike most gene expression methods in considering base-level gene expression as opposed to summary gene expression or expression of exons or splice junctions, modifications of existing normalization and variance stabilization methods are required. We perform additional pre-processing that can be thought of as determining the residual per-base expression for each sample after subtraction of a model representing the shape of that gene across all samples (Fig. 1b). We refer to a potential abnormal residual structure as latent gene expression because it is not directly observed but inferred after subtraction of the model from the normalized data.

**The proposed model**. The curve after the normalization (Fig. 1b) can be viewed as a single point in a high-dimensional space by considering each base position as a dimension and the height of latent gene expression as a value at that position. By considering each curve in the normalized data as a random vector in a high-dimensional space, we fit multiple unknown mixture distributions, recasting the problem into a high-dimensional latent variables framework (see "Methods" section). A latent variable is used to model an underlying abnormal gene expression trajectory, i.e., an outlying direction in a high-dimensional space, that is interrogated for outliers. An outlier case with shape changes then can be a data point that is strongly involved in one or multiple abnormal trajectories, which enables modeling complex structural variation.

**Shape change detection**. SCISSOR extracts latent gene expression associated with abnormal sequencing coverage and quantifies the level of abnormality in a robust way for determining the cases with shape changes (Fig. 1c, d). As the type of structure of interest here is outlying/abnormal, we search for the best one-dimensional projection in which each sample achieves the maximum deviation from all other samples, as a result, producing a direction we termed the "most outlying direction" (MOD). Recognizing that RNA-seq data may retain the quality of skewness despite normalization, we propose a modified projection outlyingness approach which is complementary to a robust measure of how outlying a sample is in the most extreme one-dimensional direction[23,24] (see "Methods" section and Supplementary Note 5). This modified approach helps to avoid spurious outliers due to strong skewness of distributions as well as enables approximate comparisons of outlyingness across genes and samples. At each gene under consideration, the resulting statistic is an outlyingness score for each sample with larger values indicating more severe deviation from other samples in the data set (Fig. 1c). The observed distribution of the outlyingness scores can be used for identifying and ranking outliers and modeling statistical significance. For each outlier, SCISSOR produces the MOD as a single best trajectory that describes abnormalities of the corresponding outlier, which can be used to recover the latent space of underlying outlier directions (see "Methods" section and Supplementary Fig. 1).

The current method increases the dimensionality of the data by considering expression at a per-base level along the entire length of the gene under consideration. Although this allows high-resolution views of RNA variation, it also entails a higher level of noise due to sampling variation. Accordingly, our approach must

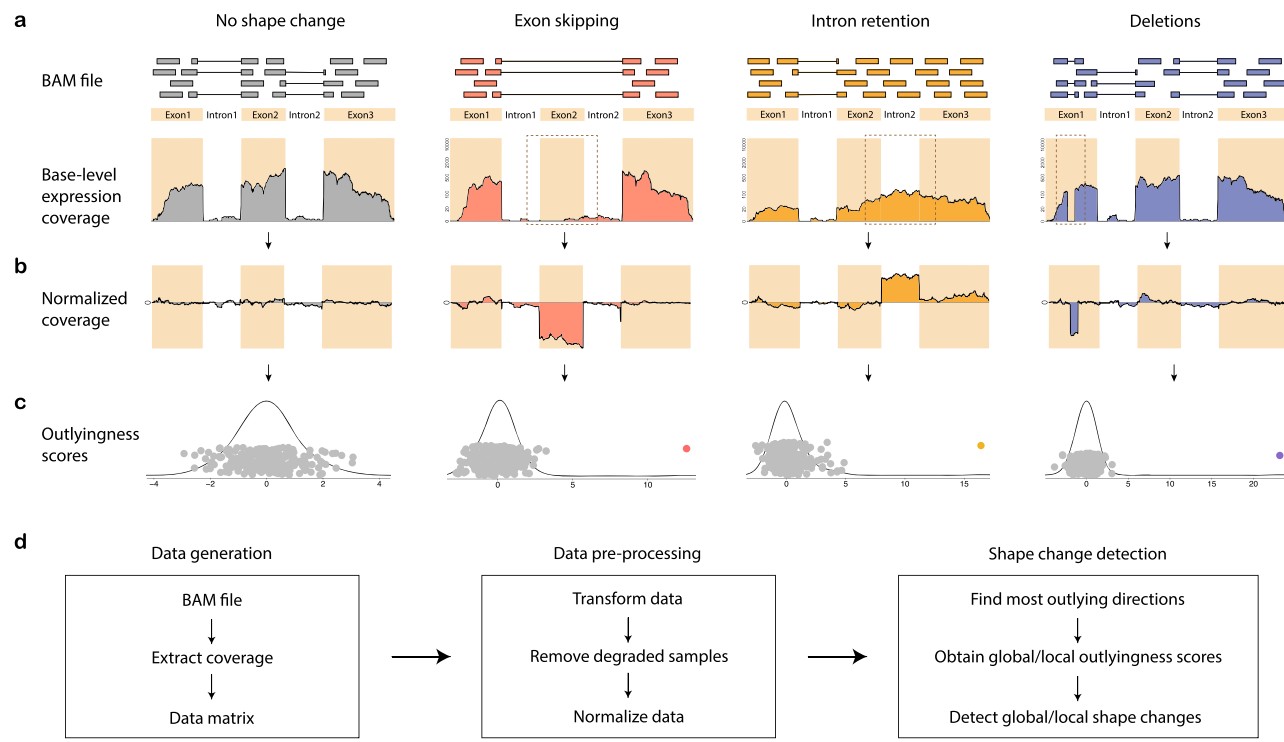

**Fig. 1 Pipeline of SCISSOR for detecting shape changes at a single gene. a** Main steps of SCISSOR are illustrated with four typical shape change examples using a toy gene: an intact example (gray); exon skipping (pink); intron retention (yellow); deletions (blue). The toy gene consists of three exons highlighted by colored background and two introns by a white background. In each example, short reads are represented by colored bars and the bars joined by the solid lines indicate the bridging reads spanning two separate genomic regions. Abnormally spliced reads from each scenario result in aberrant shapes in base-level RNA-seq expression profile as indicated by brown dashed boxes. In each coverage figure, the *x* axis represents genomic coordinates of the gene and the *y* axis represents the read-depth, i.e., the number of reads aligned to each nucleotide. **b** The normalized coverage accentuates the aberrant features by eliminating the common structure shared by the majority of samples. **c** The scores obtained by projecting the normalized coverage matrix onto each of the most outlying directions corresponding to each example are shown with the kernel density estimates. The colored point in each scatter plot indicates the outlyingness score corresponding to each example. **d** Main steps in the pipeline are outlined. RNA-seq coverage for a single gene is extracted from a BAM file for each subject, and a data matrix is constructed by collecting coverage data from all subjects. The data matrix is then pre-processed through transformation, exclusion of degraded samples, and normalization. The proposed statistical procedure is applied to the normalized data, providing the most outlying directions, outlyingness statistics, and identified shape outliers.

address the signal-to-noise ratios generated by accumulated noise in such a high-dimensional framework. Specifically, we attempt to detect with similar success, both focal and broad structural alterations in genes expressed either at very high/very low levels, as well as for very short/very long regions, challenges which we feel are not well captured by current methods.

To address these challenges, SCISSOR implements a two-step procedure, global and local, taking advantage of low-dimensional transformations and sparsity (see "Methods" section). We define a global shape change (GSC) as one that appears in a wide range within a gene, including altered exons or introns, intragenic deletions, and fusions. The directions supporting disjoint exonic or intronic regions are considered as a set of orthogonal bases of a low-rank space that will be searched by the modified projection outlyingness method. As a result, the MOD of an outlier detected by the global shape change detection is established by a linear combination of the bases (or a basis itself), which enables the interpretation of the abnormality and the automatic character-ization of its type (see "Methods" section and Supplementary Fig. 1a–c, Supplementary Table 1). By contrast, the local shape change (LSC) detection procedure only considers changes in closely related base positions. In this second step designed to detect more focal sequence changes often missing from the low-rank space, we interrogate the remaining residuals. We optimize the residual outlyingness with respect to sparse directions supporting important regions within a given gene. This helps

detect local genomic variation such as abnormal gains or losses at narrow regions (Supplementary Fig. 1d–e).

**Validation.** To validate the assumptions of SCISSOR, we inves-tigated the results applied to genes commonly altered in HNSC, including *TP53*, *CDKN2A*, and *FAT1* (Fig. 2 and Supplementary Figs. 1–8). For the purposes of this exercise, we relied on publicly reported mutation calls from TCGA as well as TCGA RNA-seq data. Although these genes are frequently mutated, overall no single structural alteration is recurrent or overlapping. As such, we provide evidence that the method detects non-recurrent out-lier events that we extend to passenger genes and infrequently mutated genes (Supplementary Note 6).

In 452 samples, SCISSOR identified 26 GSCs and 24 LSCs capturing a rich set of known (82%) and previously unrecognized (18%) genetic aberrations in *TP53* (Fig. 2a, b and Supplementary Figs. 2–4). As expected, many of the shape changes coincided with mutations (82%), most commonly at splice sites that caused exon skipping or intron retention and in-frame indels character-ized by LSCs. Somewhat unexpectedly, only 59% of mutations annotated as splice site mutations in the public repository of HNSC mutations were associated with a statistically significant shape change. Since we expect canonical splice-site mutations to result in a shape change associated with either retained intron or skipped exon, we investigated these discordant findings further. In nearly every case, we found that the called mutation was likely

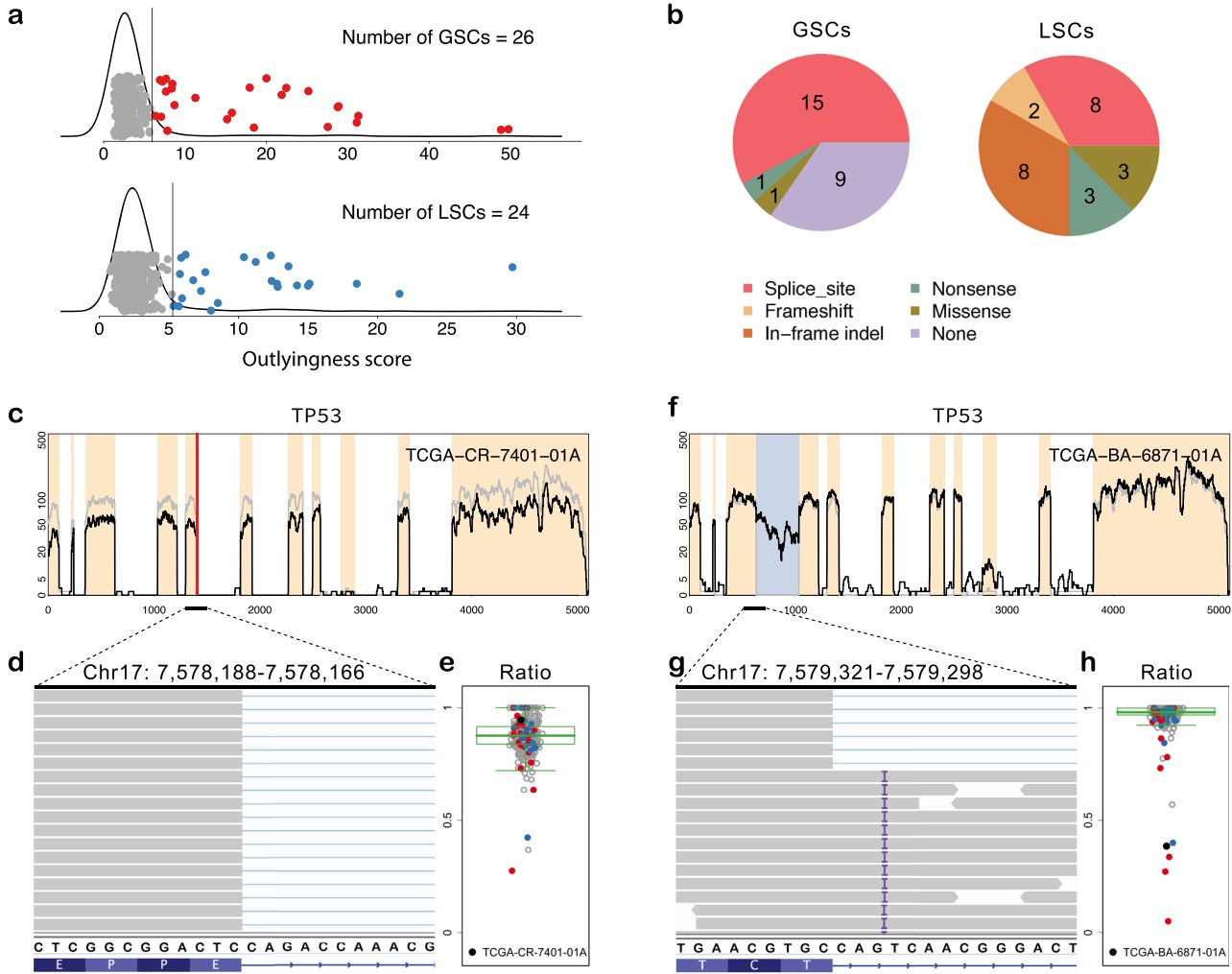

**Fig. 2 SCISSOR results at the gene *TP53*. a** Global outlyingness scores and local outlyingness scores are shown in the top and bottom figures, respectively, with kernel density estimates. The colored points indicate the identified shape changes from each step based on the cutoff values (significance level = 1e −04) indicated by the black vertical lines. Each procedure identified 26 global shape changes (GSCs) and 24 local shape changes (LSCs). **b** Association between mutations and shape changes are summarized. The numbers of coincidences of each mutation and identified GSCs and LSCs are shown in the pie charts ($n = 452$). **c** Coverage of a sample with a normal splicing pattern in spite of a splice site mutation is shown (black curve) with median coverage (gray curve). **d** The reads aligned to the exon-intron junction involved in the splice site mutation are all normally spliced and a subset of those reads are shown using the Integrative Genomic Viewer (IGV). **e** The ratio of normally spliced reads to reads aligned to the last base of the exon is shown for all samples ($n = 452$; gray for nonoutliers; red for GSCs; blue for LSCs) using a box plot with the median, interquartile range (IQR), and 1.5×IQR distances from the upper and lower ends of the box marked. The particular sample (black point) has a large ratio close to 1, indicating normal splicing. **f** Coverage of one of the identified shape changes in the absence of any mutation is shown (black). **g** A subset of the aligned reads to the region of intron retention is shown using the IGV, which appears as an insertion (T) at 7,579,308–7,579,309 in all the reads abnormally retained. **h** The ratios of normally spliced reads at this region are shown using a box plot with the median, interquartile range (IQR), and 1.5×IQR distances from the upper and lower ends of the box marked ($n = 452$; gray for nonoutliers; red for GSCs; blue for LSCs). The box plot shows that the sample (TCGA-BA-6871-01A) and a few other samples identified by SCISSOR have a large proportion of abnormal reads.

incorrectly annotated or had limited (or no) supporting reads for an abnormal splicing event (Fig. 2c–e and Supplementary Table 2, Supplementary Note 2).

In addition to shape variation associated with previously reported mutations at *TP53*, we identified 11 samples that were not documented in the TCGA mutation calls. In these 11 samples, we detected intron retention and skipping of multiple exons (Supplementary Fig. 4). In one case (but often observed in other genes), the examination of the DNA alignments in the immediate vicinity (often within 3–5 base pairs (bp)) of the shape change revealed a variant not annotated as a canonical splice site mutation, but likely functioning as an abnormal splice donor or acceptor. For example, we identified a novel intronic insertion

associated with the retained 3rd intron in TCGA-BA-6871-01A (Fig. 2f–h). Although the insertion was located 3–4 bp from the exon-intron junction which was not annotated as a splice site in the TCGA, all 38 intronic RNA reads contained the insertion whereas the exonic VAF was 0% ($P < 0.001$, Fisher's exact test). This supports a potential impact of intronic variants (>2 bp) on abnormal splicing, which has been relatively underappreciated compared to SNVs in exons and canonical splice sites[25], and thus shows the capability of our shape-based analysis to assess the functional impact of variants outside of the canonical splice sites.

Further, SCISSOR detected all of the previously reported as well as the unrecognized tumor-suppressor loss of function mutations in this disease, increasing the numbers of variant cases

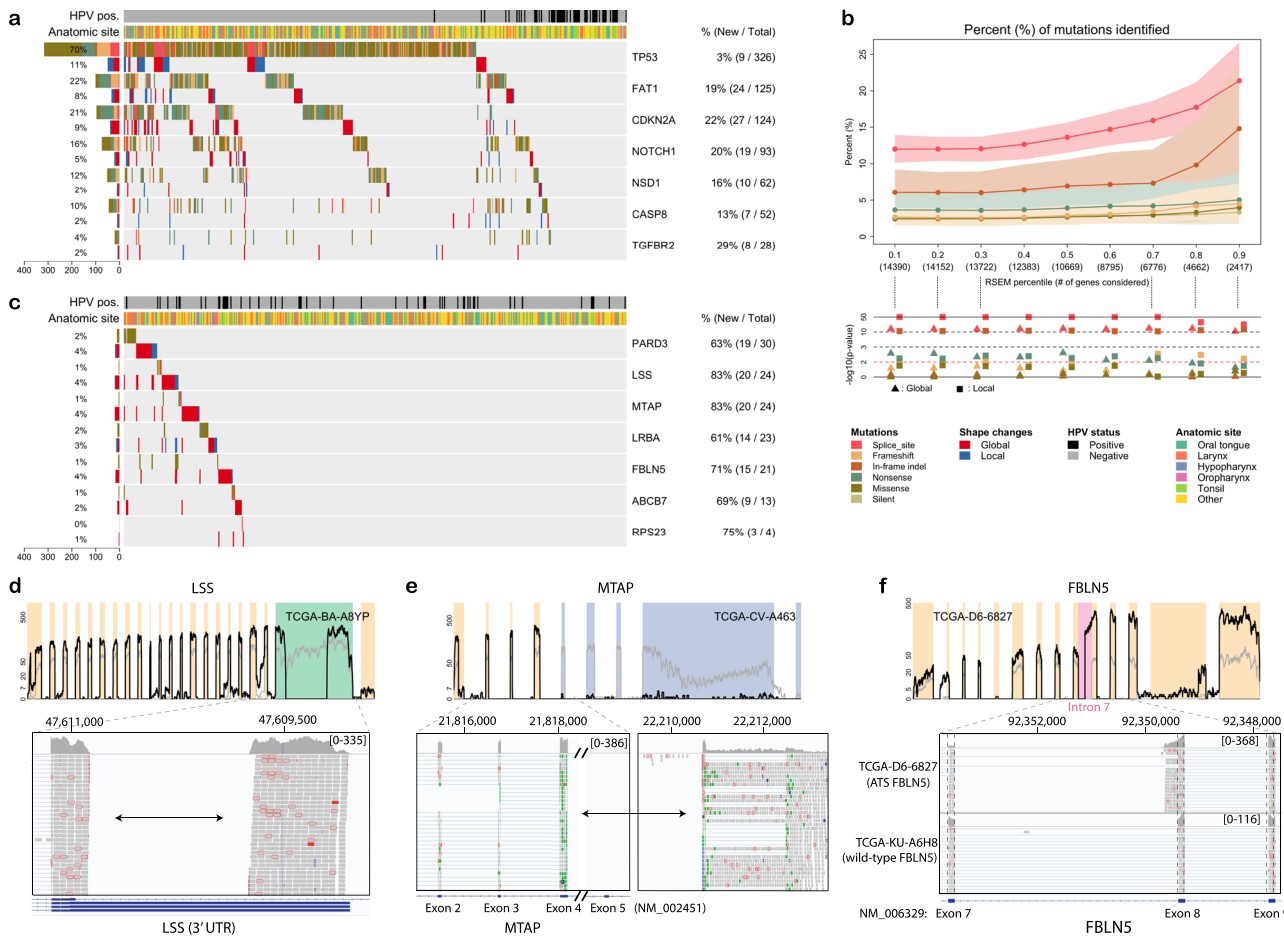

**Fig. 3 Genome-wide analysis of SCISSOR. a** Significantly mutated genes (rows) for 452 head and neck squamous cell carcinomas (HNSC) samples (columns). Each gene has two rows: mutations and shape changes. Left, percentages of mutation and shape variants; right, percentage of variants additionally identified by SCISSOR/percentage of total variants from mutation and shape changes. **b** Top: percentage of mutations identified across genes ($n = 14{,}390$; on/off genes excluded). The x axis represents thresholds for filtering out lowly expressed genes with the number of genes included at each threshold. Each threshold indicates each percentile cutoff for both median and median absolute deviation (MAD) of normalized RSEM values across samples. The error envelopes were constructed by bootstrapping samples of size 200 with 500 repeats (5th, 95th percentile). Low: as a negative control for this experiment, we report log10(P-values) from a two-sided Fisher's exact test using silent mutations as a negative control. P-value of 0.01 marked by a red dashed line. **c** Some of the novel genes identified are shown using the consistent format with **a**. **d** Recurrent abnormal expression changes in 3′ UTR in *LSS*. The RNA-seq profile, for example (TCGA-BA-A8YP), shows deep deletion in the middle of the 3′ UTR (green background). The detailed view of the 3′ UTR by the Integrative Genomic Viewer (IGV) shows no spliced junction reads. However, read pairs were aligned to each of the split regions indicated by the arrow, suggesting an aberrant splicing event. **e** Recurrent structural variants for *MTAP*. For TCGA-CV-A463, the last few exons of *MTAP* were deleted (blue background) and the remaining exons were connected to other parts of the genome. **f** Recurrent alternative transcription start (ATS) for *FBLN5*. The detailed view of *FBLN5* exons 7–9 including the intervening introns is illustrated by the IGV for *FBLN5* ATS event (top) and *FBLN5* wild type (bottom). The ATS event shows the short reads aligning continuously between intron 7 and exon 8. On the other hand, the wild type shows normally spliced junction reads between exons 7 and 8 with no expression in intron 7. Exons 7–9 are expressed at similar levels.

with statistical support between 20 and 30% on average (Fig. 3a). Specifically, variant shape changes were identified in 8% and 9% of samples at *FAT1* and *CDKN2A*, respectively, including all nine splice site mutations called at *CDKN2A* and multiple known and novel fusion-like structural variants events in *FAT1* (Supplementary Figs. 5–8). Additionally, as an example of gain of function events, SCISSOR identified shape changes associated with previously reported *FGFR3–TACC3* fusions in two samples (Supplementary Fig. 9)[21]. Taken together, we conclude that SCISSOR effectively detects altered expression not only caused by truncating mutations but also various sources such as alternate splice isoforms, fusions, and intragenic deletions.

**Genome-wide analysis**. Having investigated the use of SCISSOR in genes known to be important in HNSC, we next applied it

genome-wide. Importantly, this analysis emphasizes non-recurrent outlier events in both known driver (Fig. 3a) and potential driver (Fig. 3b) versus potential passenger genes. The distribution of the number of detected outliers was interrogated on a per gene basis in 14,399 expressed genes (after gene filtering) from the genome-wide analysis in the TCGA HNSC cohort (Supplementary Note 6). About 32% of the genes had at least one significant shape change, and many of them had only one or two shape changes identified. Only 2.7% of genes had alterations in >1% of the cohort. Overlapping TCGA mutational data with SCISSOR outliers showed a strong association between shape changes and mutations predicted to change the shape of a transcript such as splice site mutations and indels. When considering genes expressed at higher levels, splice site/indel mutations were more likely to be associated with a shape change than genes with lower expression (Fig. 3b), e.g., if a gene is not expressed, a splice

site mutation cannot alter the shape of its coverage. As silent mutations were not predicted to change the transcript structure, we performed Fisher's exact test for different mutational types with silent mutations as a negative control for this experiment. Likewise, missense mutations would not be expected to strongly impact the shape of the pileup, and accordingly, there is no significant association with shape changes.

In order to make novel observations, we investigated outlyingness scores for every gene and sample. Using the normalized outlyingness scores across samples and genes allows for prioritization despite the very high dimensionality of the genome-wide scan, which is a powerful property of our approach. We captured known driver genes including *TP53*, *FAT1*, *CDKN2A*, and *NOTCH1* but also found a number of previously unrecognized genes (Fig. 3a, c–f and Supplementary Fig. 10). For example, the gene *LSS* demonstrates a recurrent 1.13 kb deletion in the 3' UTR in nine samples (2%) (Fig. 3d). This was a notable example because no spliced reads were observed flanking the deletion in TCGA data, making the event impossible to find with existing splice-based algorithms. The Genome Aggregation Database (gnomAD) reported this event with the allele frequency of 0.05 in the general population[26], suggesting that the *LSS* shape change was driven by a common, but poorly characterized, germline CNV. Further investigation on the enrichment of this variant in our cancer cohort over the population revealed that this variant was strongly associated with African ancestry ($P < 1e-07$, Fisher's exact test). Accounting for ethnicity, there was no clear enrichment in the TCGA HNSC cohort. Functional characterization in the UTR variants is challenging, and as such, they are generally ignored or given low priority. Although the functional impact of this variant is speculative, we note that the deleted region includes a known enhancer element (Supplementary Fig. 11). We suggest that SCISSOR may be uniquely placed to discover recurrent variants in the untranslated regions of genes allowing better prioritization for function follow-up.

We considered a second example, the gene *MTAP*, a polyamine metabolism enzyme, located immediately adjacent to the commonly mutated *CDKN2A* (Fig. 3e). Previous studies have suggested it as an independent driver gene in multiple cancer types, yet it is to our knowledge never highlighted as altered in mutational or copy number analysis, likely obscured by its close spatial association with *CDKN2A*[27,28]. However, SCISSOR identified recurrent structural alterations in 12 samples (2.7%) as the first objective evidence that this gene is specifically and recurrently targeted for inactivation in TCGA HNSC. We conclude that SCISSOR can augment copy number analysis in defining the driver targets of deletion events (Supplementary Fig. 12).

As proof of another important usage of shape changes, we identified recurrent alternative transcription start/termination (ATS/ATT) in a number of genes. ATS/ATT has been reported as the principal drivers of isoform diversity and a novel mechanism of oncogene activation in preliminary reports[6,29]. However, methods are limited to supervised differential ATS/ATT detection between two or more groups or only limited types of ATS/ATT[6,29]. In contrast, SCISSOR enables unbiased characterization of ATS/ATT for individual samples by their distinct RNA-seq shape patterns which contain exons followed by ATS site at a higher level than the preceding exons, and vice versa for ATT. For example, we identified a novel *FBLN5* isoform that is expressed in 13 samples (~3%), which initiate from exon 8 preceded by ~300 bp of intron 7 (Fig. 3f). Although the functional impact is unknown, this isoform is not expressed in normal samples ($n = 44$), no supporting transcripts have been reported in a public database, and no recurrent mutations were found.

In summary, it is clear that SCISSOR identifies both novel driver genes and novel mechanisms of gene alterations not easily identifiable by any other single method.

**Concordant outliers across multiple samples in shared genes in association with internal exon CpG loci.** Having validated SCISSOR for the detection of known and novel variants in single genes, we next asked if there was evidence for shared shape changes in groups of genes across samples. We hypothesized that correlated shape changes might be detected in a number of circumstances including shared alterations in pathways, such as loss of function in splicing factor genes[30,31], shared artifacts, such as global RNA degradation[32–34], differential sample cellular composition, such as the proportion of infiltrating immune cells[8,35], or some other novel mechanism. Starting with an unsupervised approach, we investigated potential non-random associations between altered genes across samples by Fisher's exact test applied to the number of co-altered genes in every pair of the cohorts. Four subjects were identified with high significance by sharing alterations in a number of genes (Fig. 4a). Among 2417 genes that were highly expressed, ~25% were co-altered in these four samples. Manual inspection revealed that overwhelmingly these samples were identified as the result of related abnormalities in the co-altered genes, often characterized by either different usage of 5' exon or intragenic dips in coverage at or near regions of internal exon promoters (Fig. 4b). We considered numerous potential etiologies for shared variant transcript patterns across samples (Supplementary Table 3). An *NSD1* nonsense mutation known to be associated with chromatin modification was observed in one out of the four samples such as that previously reported in HNSC[36], but a consistent pattern for other *NSD1* mutant samples was not observed. Intriguingly, we observed a close localization, in nearly every case, to intragenic CpG islands, implicating a potential underlying epigenetic regulation as has been described in the context of paired DNA-histone epigenetic regulation of internal exon promoters (Fig. 4b and Supplementary Fig. 13a)[3].

Further inspection of reads associated with the shape change revealed that the impacted samples demonstrated a discernable break in the continuous transcription into two separate discontinuous transcripts as noted by the absence of either spanning or bridging reads mapped to that locus (Supplementary Fig. 13b). In the majority of cases examined, there was a known ATS site mapped to that location, and the gene was known to be alternatively spliced (Supplementary Fig. 13a, b). In fact, a targeted review of all samples revealed additional samples expressing similar variant transcripts although the variation was relatively weak (Supplementary Fig. 13c). We believe that variant gene transcription of this type would be difficult or impossible to detect by other existing methodologies. The adjacent exon coverage was similar such that exon-based detection would not identify them. Additionally, since the events often occur internally to exons and are characterized by the lack of expression rather than differential splicing, alternative splice approaches would likewise fail to detect them.

**Evaluation.** There are challenges in comparing SCISSOR to the many transcript analysis techniques which largely focus on supervised analysis with the goal of detecting differential isoform usage between sample groups[13–17,37]. Nonetheless, we endeavored to compare any potential benefits of the high-dimensional transformation of RNA offered by SCISSOR to approaches that rely more on lower dimensions such as exon summary data or alternative splicing[13,25]. We adapted methods representative of junction reads-based analysis (JRBA) and exon-level expression-based

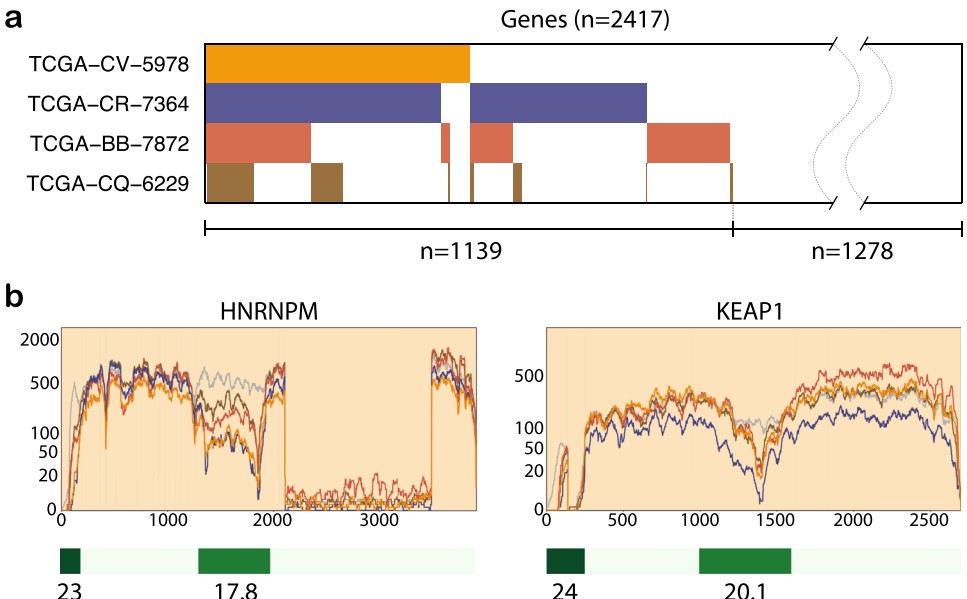

**Fig. 4 Concordant shape changes and CpG islands. a** SCISSOR identified four samples (0.9%) that showed concordant shape changes in a number of genes. The heatmap shows the highly expressed genes ($n = 2417$) in columns and the genes where each of the four samples is altered are colored in each row. About 30% of the highly expressed genes were co-altered in at least two of the four samples. **b** To illustrate the concordant shape changes, the RNA-seq profile of the four samples is shown in two genes, *HNRNPM* and *KEAP1*. These samples demonstrate two common alterations, either alternative usage of 5′ exon or intragenic dips in coverage at or near CpG islands. Below, the CpG islands are indicated by the green color codes with the percentage of GC contents.

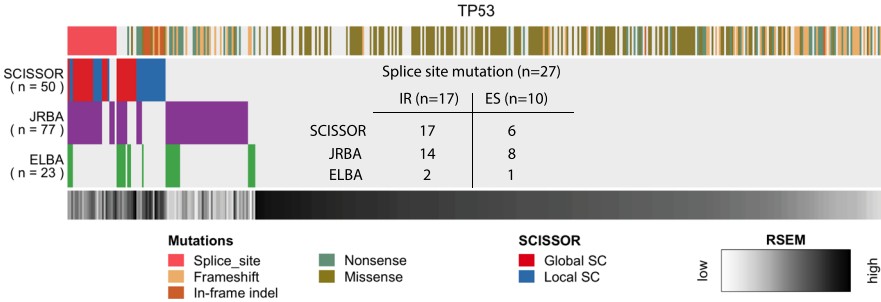

**Fig. 5 Comparison of SCISSOR, JRBA, and ELBA at *TP53*.** The outliers identified by SCISSOR, JRBA (junction reads-based analysis), and ELBA (exon-level expression-based analysis) are shown. Color codes for mutations and gene expression are shown on the top and bottom of the heatmap, respectively. In the color code for mutations, the splice site mutations disturbing splicing ($n = 27$) are all shown on the left side (pink). Many of these splice site mutations were identified from both SCISSOR and JRBA. The table in the middle of the heatmap summarizes the number of splice site mutations identified by each method in two separate cases: intron retention (IR) and exon skipping (ES). Many of the additional shape changes (SCs) from SCISSOR are associated with deletions in the middle of exons that are difficult to detect by junction reads or exon-level expression. On the other hand, most of the additional outliers from JRBA or ELBA are associated with a low expression as shown in the color code for gene expression.

analysis (ELBA) such as percent-spliced-in and DEXSeq[2,13,15,30], and compared their performance in identifying the canonical alterations of *TP53* as well as in simulated data (Supplementary Note 3). We found that all three approaches detected significant abnormal exon usage characterized by whole exon skipping and multiple partial exon-skipping events, supporting the observation that SCISSOR was at least comparable to prior approaches for this variant class. However, ELBA mainly had difficulties in recognizing weaker aberrations in exon usage such as the cases where only a limited fraction of reads skipped exon. Considering splice site mutations as a positive control, JRBA produced a comparable performance as SCISSOR by covering intron retention events whereas ELBA generally failed to detect this class of alterations (Fig. 5). Compared to JRBA, SCISSOR is most challenged when the variant splice results in an acceptor splice site that is only different by a few bases such that the global shape of the gene is not severely perturbed (Supplementary Note 3). By contrast,

SCISSOR identifies a set of samples that is primarily characterized by smaller indels that are not recognized by either ELBA or JRBA. Additionally, SCISSOR outperforms competing methods when the alterations are in complex transcript variants that do not result in obvious abnormalities in splice counts or gene/exon-level expression (e.g., Figs. 3e and 4b).

We observed that ~50% of the variants identified by JRBA were unique to that method and corresponded to samples with overall low expression by RSEM. Although a definitive characterization is challenging, we suspect these are false positives. The simulation studies further showed that SCISSOR outperformed JRBA at every coverage level in the sensitivity as well as the false discovery rate (Supplementary Note 3). Similar to JRBA, approximately 50% of the samples identified by ELBA might have been more accurately described as absent gene expression overall. We conclude that the identification of large numbers of false positives at low gene expression levels is a significant limitation of

exon-based and splice junction based approaches[18]. In contrast, SCISSOR produced unbiased results, less dependent on gene expression levels, thanks to its normalization procedure which helped to minimize trivial and unreliable signals from low coverage. A detailed discussion for the comparison is provided in Supplementary Note 3.

**Filtering genes and samples**. Here, we consider sample and gene filtering required for the efficient use of SCISSOR. The SCISSOR methodology can be impacted by cases with degraded RNA since degraded transcripts will register as outliers systematically showing increasing slope from 5′ to 3′ across many genes[32–34]. For 522 HNSC tumor patients, we measured the amount of degradation across genes and identified 70 cases that were severely degraded in a large fraction of genes and excluded from the downstream analysis (Supplementary Fig. 14 and see "Methods" section). The analysis did reveal one previously unappreciated fact, which is that degradation did not appear uniformly across all genes but differentially impacted longer transcripts (Supplementary Fig. 14b, c). Specifically, we noted in the TCGA data set, transcripts under 2500 bases rarely demonstrated steep slopes even in cases with significant evidence of RNA degradation overall (Supplementary Fig. 14b). For gene filtering, a gene that is not expressed in some or all samples violates the data distribution assumptions of the method. We note that SCISSOR calculates latent gene expression from a model of that gene's expression across all samples in the cohort. The interpretation of latent gene expression from a gene that is itself not expressed in many or most samples requires special considerations that are not addressed in the current work. A per gene definition of "not expressed" versus "expressed" was developed for the purposes of excluding genes with many non-expressed samples for a given gene (Supplementary Fig. 15a). As a result, we identified 5802 genes (~28%) and they were excluded from the downstream analysis (Supplementary Fig. 15b).

## Discussion

Here, we report a robust method, SCISSOR, that considers a shape property of aligned short read data through a transformed pileup file. With the goal of detecting samples exhibiting anomalous shapes, it models base-level read counts using a high-dimensional latent variable framework that is naturally integrated into its normalization, abnormal feature extraction, and quantification. As a result, it offers a comprehensive and computationally efficient tool that identifies a range of genetic alterations including abnormal splicing, ATS/ATT, and small or large deletions.

Our work clarifies the importance of many variants by either confirming their function, or in many cases questioning their relevance. In addition, we provide motivating examples of recurrent mutations in the UTR of genes, internal exon splicing, and cryptic events such as large intragenic deletions that would be difficult to characterize by most other methods. Relative to other approaches, which tend to identify thousands of significant events at the cohort level that are difficult to prioritize, SCISSOR independently prioritizes genes with a known role and identifies with high confidence novel targets for efficient review. SCISSOR has also suggested that internal exon DNA methylation may be a common source of alternative transcription in HNSC. Finally, SCISSOR provides a novel approach to consider not only outlier shape changes, such as disrupted transcription and fusion events, but also systematic shape changes such as degradation.

Although we have shown that consideration of base-level coverage offers the potential to detect known and novel classes of variants, there remain several challenges. Systematic coverage bias

may occur within a single sample or batch of a cohort due to RNA degradation, fragment length, or other unknown factors. We have shown the ability to detect samples with degraded RNA across genes in specific samples. After removing such samples, we then attempted to detect residual RNA degradation in calls of outlier genes as assessed by clustering outlier genes and samples. No evidence of residual RNA degradation was observed, although we recognized that our inability to detect it does not exclude the possibility of its existence. Likewise, although TCGA samples such as we have used have high-quality RNA, it is likely that factors such as variable fragment length would introduce artifactual signals detectable by SCISSOR. The impact of GC content on nucleic acid sequencing is well known to reduce template amplification, and as such introduce artifactual shape changes. When such changes are systematic across all samples, they will be removed by normalization, but if experimental conditions such as temperature are varied, we might observe sample-specific effects as well. To the extent that these effects are measurable by SCISSOR, they will complicate the biologic interpretation of any results.

In the current work, we describe a method for detecting outlier RNA events across samples. Changes may be due to physiologic events, such as alternate splicing, pathologic events, such as mutation, or experimental factors such as RNA degradation or contaminating stromal cells. To the extent that outlier shapes are detected by SCISSOR as represented by the underlying data, we consider this a success of the method, even when those changes might relate to experimental factors such as RNA degradation (clearly seen in the data), contaminating stroma cells (likely observed in the current data set and the source of future reports), GC content (potentially seen in the current report as described), or RNA fragment length (not obviously detected in the current report but potentially a concern). Users of SCISSOR should consider experimental as well as biological factors when interpreting the results of SCISSOR. In this work, we observed that RNA degradation, if not addressed, will result in degraded samples producing large numbers of outlier genes. We addressed this by empirically removing samples with evidence of degradation. We observed some weaker evidence coordinated RNA shapes shared by samples across many genes in association with certain types of internal CpG islands. Whether or not this is a GC experimental artifact or a biologic process is unclear from the current data. For those samples with evidence of retained intron, we considered the possibility of DNA contamination as an explanation. DNA contamination as assessed by the 260/280 ratio was consistent with pure RNA. Additionally, we would expect that contaminating DNA would be evidenced as a general phenomenon across many introns at lower or consistent levels rather than as we see it in highly selected introns at high coverage in a very limited number of introns.

In its current form, SCISSOR has tuning parameters including log normalization and a normality cutoff value which will likely require some optimization when a significantly different wet lab RNA-seq protocol is used. In addition, because the optimization performed in SCISSOR is intensive and even infeasible for high-dimensional data, SCISSOR provides a good compromise with dimensionality and sparsity by implementing a two-step procedure. Possible issues from this are that the GSC detection algorithm loses information about orders of base pairs and that the LSC detection algorithm might give alternative results with different window sizes. For a future study, a complementary approach would be an all-in-one procedure that maximizes the outlyingness with respect to full-dimensional window directions by simultaneously searching for the optimum location and size of a window for each sample. Although our attention was focused on RNA-seq data, SCISSOR has great potential for future

applications, including detection of non-coding RNA, DNA shape changes such as copy number alterations, and single-cell RNA-seq data. Such applications will likely explore some of the tuning parameters and other optimization in more detail.

## Methods

**Modeling base-level read counts**. At a given gene, SCISSOR starts with a read count matrix $R$ each entry of which is the observed read count $R_{ij}$ mapped to each base position $i$ in each sample $j$. The dimension of the matrix $R$ is the number of base positions along the gene locus which is denoted by $d$. Therefore, the input pileup data $R$ we consider at each gene is a $d \times n$ matrix in which $d$ indicates the length of the gene, i.e., dimension, and $n$ is the size of the cohort. In other words, each sample is a $d$-dimensional read count vector which can be thought of as a $d$-length trajectory with certain shape patterns (Fig. 1a).

We model read counts $R_{ij}$ as following multiplicative framework in a high-dimensional setting:

$$R_{ij} \approx \mu_i^{a_j} \cdot m_{ij} \tag{1}$$

with the geometrical mean $\mu_I > 0$, scaled by a normalization factor $a_j$, and the other sources of variation $m_{ij}$ including alternative shape variants.

The proposed model is divided into two parts. The first part is $a_j$, which represents differences due to sequencing depth and serves as a sample-specific scaling factor. We also model a small additional bias remaining in a nonlinear manner with respect to the $a_j$, possibly because of gene-specific variation including GC content, mapping bias, and other nonlinear factors, using a smooth function $g$ (.). In many genes, we observed certain variations non-linearly associated with the overall expression ($a_j$) in a gene-specific way (Supplementary Fig. 16). For example, in some genes, highly expressed samples tend to have more noisy variation along the locus compared to moderately expressed samples. Such unequal noise levels can confound the downstream outlier detection procedure, so our normalization step adjusts this nonlinear variation by estimating $g$(.) as a smooth function of an overall expression level satisfying a constraint $g(1) = 1$. Using this $g$(.), the second part of the model, $m_{ij}$, for remaining variation after adjusting for sample-specific expression levels, is further modeled by $\log\left(m_{ij}\right) = g\left(a_j\right) x_{ij}$ with $d$-dimensional random vectors $X \sim P_d$, where $X_j = \left(x_{1j}, \ldots, x_{dj}\right)^T$. In this way, we used the variable $a_j$ to account for a linear ($\mu_i^{a_j}$) as well as nonlinear ($g(a_j)$) effects of systematic variation that should be corrected. With this representation, SCISSOR provides a natural way to decouple outlier-associated signals from systematic variation by estimating and normalizing out the $a_j$-associated terms from raw read counts (Fig. 1b, see "Methods" section and Supplementary Note 1).

The random vectors $X_j$'s describe other sources of variation beyond overall expression levels, including diverse genetic events that may lead to shape changes in expression. To model these variations, we represent the vector $X_j$ by a spiked covariance model[38–41] with a set of underlying signal directions $\{U_i\}_{1 \le i \le K}$. These signal directions include latent outlier directions potentially generating outliers, and to account for the underlying variation associated with outliers, we model $X_j$'s as follows:

$$X_j = \sum_{i=1}^{K} y_{ij} U_i + \epsilon_j, \text{ where } y_{ij} \sim \begin{cases} \sqrt{\tau_{i,1}} z_{ij}, & \text{w.p. } 1 - w_i \\ \sqrt{\tau_{i,2}} z_{ij}, & \text{w.p. } w_i \end{cases} \tag{2}$$

where the $z_{ij}$'s are assumed to be i.i.d. random variables with mean zero and variance one. For each $i$, by assuming a small $w_i$ to be an unknown proportion of outliers associated with the $U_i$ and $\tau_{i,2} \gg \tau_{i,1} > 0$, the mixture distribution reflects the underlying mechanism generating an outlier that goes strongly in the direction $U_i$. The model also allows an outlier to be associated with several components, which offers flexibility in modeling more complex RNA-seq outliers that show multiple aberrations. As such, a sample vector from Eq. 2 can be viewed as a high-dimensional random vector from a complicated mixture distribution whose components have different covariance structures.

**Normalization**. The normalization step aims to reduce unwanted technical bias and obtain the fundamental variations that are essentially associated with outlying signals. Based on the proposed model Eq. 1, this can be naturally done by decoupling $x_{ij}$ from the $a_j$-associated factors. The normalization procedure is divided into two steps: the variance stabilization for finding a proper log-transformation and the mean scale correction for estimating unknown parameters and finally taking $x_{ij}$ apart from $R_{ij}$.

*Variance stabilization*. The raw coverage data often show heterogeneous variation typified by extreme skewness and different fluctuations within and between samples, implicating that highly expressed regions may dominate other biologically important variation. A typical remedy to adjust such heterogeneity of counts data is log-transformation[42,43], which helps to get more stable variation. It is common to add a shift parameter, also known as a pseudo-count, before the log-transformation to avoid the undefined zone of the log function and to control unwanted biases from low versus high. However, there is no consensus on what value of a shift

parameter should be used. For the purpose of mining outliers, it is important for the data distribution to have a roughly symmetric distribution to avoid confounding outliers from high skewness. It is very unlikely that there is a common value that minimizes the skewness at every base position and so we want a parameter that fits best in some overall sense. Therefore, we propose to select a parameter that minimizes the overall skewness of the data. To measure the overall skewness, we use $a_j$ in our statistical model as a representative value of the $d$ read counts along with the transcript for the $j$th sample. The estimate of the $a_j$ can be obtained by the estimate of the coefficient in the linear model, termed by a mean scale factor (MSF). See Supplementary Note 1 for details. Note that the MSFs depend on the log shift parameter. Thus, the resulting overall skewness using MSFs also depends on the parameter, and thus we can choose the parameter by comparing the overall skewness from different values of the parameter. By a grid search based on a given range for the parameter, we select the parameter that achieves the minimum skewness. The proposed algorithm is fully described in Supplementary Note 1.

*Mean scale correction*. While the log-transformation step helps to stabilize the different levels of variation and significantly reduce extreme skewness in data, it has been observed in a substantial number of genes that there remain some variations showing systematically nonlinear patterns (Supplementary Fig. 16a). In the model (Eq. 1), we introduced a smooth curve $g$(.) to account for such remaining variations that may lead to biased results. In practice, we only observe $R_{ij}$, and the other unknown elements $\mu_i$, $a_j$, and $g$(.) should be estimated. In brief, the estimation procedure follows as: (1) estimation of $\mu_i$'s based on a trimmed mean at each base position; (2) estimation of $a_j$'s by a robust linear model; (3) estimation of $g$(.) as a function of $a_j$ by fitting a smooth curve. See Supplementary Note 1 for full details. These steps are designed to robustly estimate the parameters and thus correctly capture the dependency on overall expression. Together, we obtain the normalized data $x_{ij}$ as a main data object for the downstream shape change detection analysis by ruling the irrelevant terms out using the estimates (Supplementary Fig. 16b).

**Modified projection outlyingness**. The transformation of RNA-seq to latent gene expression opens the door to many types of hypothesis testing, including detection of clusters and outliers. In the current work, we consider the testing for outliers. Specifically, SCISSOR tests the hypothesis that each sample is no farther apart in high-dimensional latent gene expression from the group than would be expected by chance at a given significance level. The alternative hypothesis is that a sample is farther apart than would be expected by chance (Fig. 1c). As analogous to a robust measure of outlyingness in 1-dimension, the projection outlyingness of $y$ with respect to the data matrix $Y$ for $p$-dimension is defined as $\max_{\|h\|=1} \left| \frac{h^T y - \text{Med}(h^T Y)}{\text{MAD}(h^T Y)} \right|$. The projection outlyingness finds the maximum outlyingness of a data point $y$ by looking at all one-dimensional view of a data set[23,24]. Although it is known as a robust statistic, it is most effective when the resulting distribution of each one-dimensional view is roughly symmetric. However, it is often the case that RNA-seq coverage involves strong skewness, multi-modality, or high concentration near-zero possibly due to existing clusters or a number of lowly expressed samples. Such departure from normality can produce false discoveries and conceal biologically important outliers, and therefore some attention is needed. Here, we propose a modified projection outlyingness approach taking into account departure from normality. To do this, we add a normality constraint to the projection outlyingness function so that we only search the directions in which the given constraint is satisfied. Let $\phi$ be a function measuring the normality. Then, the projection outlyingness will be

$$O(y|Y) = \max_{\|h\|=1} \left| \frac{h^T y - \text{Med}(h^T Y)}{\text{MAD}(h^T Y)} \right| s.t. \phi(h^T Y) \le \rho, h \in \mathbb{R}^p \tag{3}$$

where $\rho$ is a cutoff value. While any measure of normality can be used, here we employ the winsorized Anderson-Darling statistic with a special emphasis on the skewness[44]. This statistic is robust against outliers, allowing us to keep useful directions. From Eq. 3, we can obtain reliable outlyingness scores by looking for directions where spurious outliers are less likely associated. For the real data analysis in this paper, we used $\rho = 3$. The concept of an outlier is with respect to an expected distribution. As such any cohort to which SCISSOR is applied should be sufficiently large to assess the underlying distributions and any outlier of interest should be sufficiently large to be detected in a cohort of that size. Precise experiments to measure effect size and distributions are the subject of future work, but empirically a cohort of 30 samples would be the smallest we have considered successfully to date.

**Global shape change detection**. Extraction of latent outlier directions from a high-dimensional data set is not simple because many features are involved in an overwhelming number of dimensions. One challenge is that most solvers of this problem require that the sample size is larger than the dimension. Second, even if a solver can provide a solution for full-dimensional data, the solution might not be meaningful because of the swamping effect from high dimensions. So it is advantageous to make use of the knowledge about what types of aberrant features should be considered.

We propose a two-step procedure each step of which is designed to reveal particular types of aberration. The first step aims to find global shape changes that involve exon- or intron-level changes such as large structural variations including fusion and large deletion. To identify global shape changes, we first approximate the data matrix using a set of components (high-dimensional directions or bases) each of which spans an exon or an intron. As projection outlyingness is independent of the coordinate system chosen[24], we reconstruct the data points regarding the selected orthogonal components as new coordinate axes. With this new coordinate system, we apply Eq. 3 by taking $y = x_j$, $Y = X$, $p = K$, and obtain $n$ projection outlyingness values, denoted by $GO_j$ ($j = 1, \ldots, n$), for $n$ data points. Under the normality assumption, the $GO_j^2$ follows the chi-squared distribution with $K$ degrees of freedom (df) where $K$ is the number of basis directions included. To find a more accurate cutoff value for the case beyond the normality assumption, we implemented a data-driven approach to search what degrees of freedom fit the $GO_j^2$ the most. Starting from df = 1, we performed the Kolmogorov–Smirnov (K–S) test with the $GO_j^2$ with potential outliers excluded and looked for the df that attained the smallest K–S statistic, or the largest $p$-value. Then, we declared global shape changes using the chi-squared distribution with the chosen df based on the pre-determined level $\alpha$.

**Local shape change detection**. We now propose a second step procedure to deal with more challenging situations when outlying features are not distinguishable using the low-rank representation constructed by exon/intron bases. In RNA-seq data, it has been observed that shape variation associated with such weak signals often exhibit local changes in a limited region of base-level coverage. As these local shape changes generally remain in residuals, i.e., data after subtracting the low-rank representation, they are difficult to infer due to accumulated noise in residuals. Thus, the projection depth idea or other conventional outlier detection algorithm may be improper or infeasible.

To address this challenge, the second step adopts sparse directions supporting biologically important regions as candidates where projection outlyingness would be considered. We consider cryptic regions estimated by junction split reads as well as sequential genomic regions. Each look can be considered as a window direction which is a unit vector whose entries corresponding to a given region are all equal to a constant and the remainder of the entries are zero. The sparsity of a window direction helps to reduce the impact of noise and thus to separate meaningful outliers from inliers. Windowing approaches, although empiric, have been widely used in functional data analysis and genomics. As with all empiric strategies, the selection of input parameters can impact the results. Narrow windows can be sensitive to undesirable small fluctuation and too large windows can be vulnerable to noise accumulation. We considered windows of 50–200 for a window size based on the read lengths of short-read sequencing data. Using a collection of these window directions, denoted by $w$, we can accurately capture challenging local shape changes while reducing the impact of an overwhelming number of dimensions. The consideration of window size was validated by the performance of the method (Figs. 2b, 3b and Supplementary Note 7). To identify local outliers, we take the set of direction vectors where the projection outlyingness function will examine to be $w$. Let $I_2$ be a set of the remaining sample indices after excluding the global outliers and also let $\check{x}_j$ be the $j$th sample residual vector for $j \in I_2$. Then, the projection outlyingness for local shape changes, denoted by $LO_j$, can be obtained using Eq. 3 by taking $y = \check{x}_j$, $Y = \check{X}$, and $\mathbb{R}^p = w$, where $\check{X} = \left(\check{x}_j\right)_{j \in I_2}$. Similarly to the $GO_j$, we empirically find the degrees of freedom of the resulting distribution using the data-driven approach.

**Most outlying direction**. For a given outlier, the most outlying direction (MOD) is defined as the direction that achieves the maximum in Eq. 3 converted back to the original data-space representation. The MOD describes the individual structure of each data point that makes the point most distinguished, and accordingly, it can be used to recover latent outlying expression that possibly generated shape changes in the mixture model (Eq. 2).

The identified MOD would be some exons, introns, or their combination for a global shape change whereas it would be some narrow area, rather than an entire exon or intron, for a local shape change. Thus, it naturally enables the interpretation of the identified abnormal events. Further, SCISSOR incorporates the MOD with the splice junction reads to determine whether the event is supported by splice reads or not, providing an automatic characterization of outlier types, e.g., (cryptic) exon skipping, (cryptic) intron retention, alternative transcript initiation/termination, or small deletion, etc (Supplementary Table 1). This greatly improves the interpretability of the method and facilitates the comparison of outliers between different groups.

**Filtering out degraded samples**. It is well known that sequencing degraded RNA samples often leads to less read coverage at the 5' end of the gene and negatively affects subsequent analyses such as transcript quantification, gene expression profiles, and fusion detection[32–34]. In particular, this leaves degraded RNA-seq samples susceptible to being considered as shape outliers, which could swamp the detection of pathophysiologic aberrations in favor of alterations simply based on poor sample quality. A recent study reports that the transcript coverage of degraded samples shows

an exponential decrease as a function of the distance from the 3' end of mRNA that more highly degraded samples show a faster rate of decrease[32]. This motivates us to measure the extent of degradation, also called decay rate, by the mean-corrected slope of log-transformed RNA-seq data. To accurately assess the decay rates, we first adjusted the different sequencing depths at each locus by using the first step of the scale normalization method. This procedure helps to remove the intrinsic slopes, allowing for high-quality RNA-seq samples to be free of decreasing trend from the 3' end so that the remaining trend can be observed only in a set of degraded samples. Therefore, it enables a more accurate comparison of decay levels across genes by adjusting the other sources possibly affecting slopes. After the adjustment, we fitted a linear model to the mean-corrected coverage with the ordered base positions as a covariate for each sample. Let $q_{ij} = q_j(i)$ be the mean-corrected coverage for the $j$th observation ($1 \le j \le n = 522$) where $i$ indexes base position at a given locus ($1 \le i \le d$) of which total length is $d$. The linear model

$$q_{ij} = q_j(i) = \alpha_j + \beta_j \times \left(\frac{i}{d}\right) + \epsilon_{ij}$$

was fitted and the least square estimates $\hat{\alpha}_j$ and $\hat{\beta}_j$ were obtained. Note that we divided the covariate $i$ by $d$ to correct the effect of gene length. Then, the $\hat{\beta}_j$ is the decay rate of the $j$th observation with a higher value of $\hat{\beta}_j$ indicating severe degradation.

To identify degraded samples, we obtained $n = 522$ decay rates at each gene and collected those values across genes as a large matrix. Unsupervised hierarchical cluster analysis was performed with this matrix using hclust in R/Bioconductor with the complete linkage method (Supplementary Fig. 14a). Based on this cluster analysis, we identified 70 RNA-seq samples with strong evidence of degradation and excluded them from the downstream analysis. As expected, longer genes tend to undergo more degradation as long genes are fully affected by degradation whereas short genes are less affected. We also found that our decay rates are strongly correlated with the 3'/5' biases, an alternative measure of degradation status at two housekeeping genes (*ACTB* and *GAPDH*)[45], supporting the suitability of the proposed decay rates.

**Filtering out on/off genes**. It has been observed that the genes where samples do not share an analogous pattern commonly have a considerable number of samples that appear to be unexpressed, and we call those genes "on/off" genes (Supplementary Fig. 15). To identify on/off genes, we measure the level of shape-similarity among samples in the context of angles in a high-dimensional space between each individual vector and the mean vector at a given gene. A larger angle from the mean vector indicates that the corresponding sample presents a higher dissimilarity from the other samples. The angle approach allows a sample expressed at a low level along with the global structure that is shared among the other samples to be considered as "on" at the given gene. This helps to distinguish the signal-involved low coverage from noise. See Supplementary Note 4 for full details. We identified the 5802 on/off genes where more than 20% of the samples are off and these genes were excluded from the downstream analysis (Supplementary Fig. 15).

**Data sets**. A collection of 522 HNSC tumor samples were obtained from the TCGA Research Network[21]. The samtools was used to obtain per-base read counts. Because there often exist multiple transcripts in a given gene, mapping reads based on a single transcript may overshadow novel genetic events occurred outside the transcript. To account for this, reads were mapped to union of transcripts obtained from the TCGA generic annotation file (GAF) v2.1 based on the December 2011 version of the UCSC Gene annotations. In contrast to many RNA-seq data analyses based on reads mapped to exonic regions with splice junctions, we include reads mapped to both exonic and intronic regions in order to include potential intronic aberration. Mutations were analyzed based on mutation annotation format (MAF) obtained from the TCGA Research Network[21].

**Inclusion of intronic part**. Although SCISSOR does not require a gene model for most of its underlying assumptions, gene models greatly facilitate the biologic interpretation and visualization of results. The SCISSOR procedure, including normalization and outlier detection, can be applied directly to RNA aligned to the genome in a truly unbiased manner. However, such a procedure can produce results that are difficult to interpret as RNA is most easily described in the context of named genes, exons, introns, and splicing. As such, the SCISSOR procedure is applied in the context of named gene models to assist in the visualization and interpretation of the data. SCISSOR is unbiased in that space because it is not constrained by existing exon start or stop positions and incorporates intron space to allow detection of events outside exons such as run-on transcription. Additionally, although SCISSOR does not include splicing information in the statistical procedure, it does incorporate splices in the interpretation of shape changes once they are identified as described in the earlier section (Most outlying direction).

Gene models were modified to include a portion but not all of intronic regions to facilitate the common alterations that involve intron-exon boundaries. The omission of large portions of introns is reasonable because they complicate the visualization of RNA pileups and add little to biologic signal. The disadvantage of this decision is that important alternate splice sites and cryptic exons may be missed. To determine which parts of introns to be included in the model, a basic

rule is that the total lengths of bases for all exons and all introns at a gene to be approximately equal for the current SCISSOR application. This helps to make variations of expression at exonic regions and intronic regions comparable. Further, the lengths of intron included between exons are determined by the following rule: for each intron between exons,

1. If its length is less than or equal to a threshold ($L$), it is fully included.
2. If its length is greater than $L$, then the part of the intron is taken to be the union of two subsets of length $\frac{L}{2}$ from both ends of that intron, and the rest of it is discarded.

Here, the threshold $L$ is chosen such that the difference between total lengths of exons and introns is minimized. As a result, the RNA-seq data as a data object for the analysis consist of exons and introns with equal weights.

**Reporting summary**. Further information on research design is available in the Nature Research Reporting Summary linked to this article.

## Data availability
The RNA-seq data that support the findings of this study are available in The Cancer Genome Atlas (TCGA) database (https://gdac.broadinstitute.org/). Binary alignment (BAM) files for TCGA head and neck samples were downloaded from the TCGA Data Portal (https://portal.gdc.cancer.gov/). The mutation data were downloaded at https://gdac.broadinstitute.org/ and the gene annotation file was downloaded at https://gdc.cancer.gov/about-data/data-harmonization-and-generation/gdc-reference-files. All data are available from the corresponding author upon reasonable request.

## Code availability
An open-source software implementation of SCISSOR is available on Github[46] (https://github.com/hyochoi/SCISSOR).

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

## Acknowledgements
The research reported in this publication was supported by the National Cancer Institute under award numbers U10CA181009, CA211939, CA210988, and UG1CA233333.

## Author contributions

H.Y.C. developed the method, wrote/edited the paper. H.J., X.Z., K.A.H., S.N., J.H., M.C.H., and M.I.L. provided discussion and edited the paper. J.S.M. provided discussion and wrote/edited the paper. D.N.H. conceived and supervised the study and wrote/edited the paper. All authors performed analyses, proofread, made comments, and approved the paper.

## Competing interests

The authors declare no competing interests.
