## [Peer Review File · Nature Communications]

Reviewers' comments:

Reviewer #1 (Remarks to the Author):

In Choi et al., the authors present a novel method to infer changes in mRNA structures based on shape changes in an RNA-sequencing coverage profiling. They were able to identify small/large deletions, as well as mutations that may affect splicing. The methodological advance of this study is in the development of a very complex model to distinguish outlier-associated signals from random and systematic variation - including differences in sequencing depth and gene expression levels. I was very excited to try the method, and could clearly see how it may be useful. However, as presented, it is not clear (at least, not to me) as to what the method does intuitively (exposition), and how well it will function in a variety of experimental settings (validation).

Major comments:

1. The authors provide strong evidence that the method can pick up true signals (splice-aberrations in tumor-suppressors), but provide little evidence that the method can detect non-recurrent outlier events in passenger genes (or infrequently mutated genes) affected by mutations or structural variation.
2. It is also unclear why the authors decided to look for TP53 outliers in HNSCC given the great recurrence of TP53 mutations in this cancer type. Indeed, most samples are mutated—which, in a way, invalidates their assumption that wild-type samples are the minority. A different cohort with a lower TP53 mutation frequency may be more appropriate.
3. Clearly the authors have novel insights into the analysis of transcriptional profiles as well as were able to implement a statistical model and useful tool identify samples with unusual patterns. However, the method itself is not introduced in a way that would allow most readers to understand the intuition behind the approach. While the statistics and equations are reasonably well explained, the manuscript is riddled with difficult concepts, e.g.: "A latent variable is used to model an underlying abnormal trajectory, i.e. an outlier direction in a high-dimensional space, that is interrogated for outliers."
4. It is unclear how big the cohort-size has to be for SCISSOR to work. It also not explained whether it can be used in a case-control setting and with a limited number of replicates.
5. The introduction is lopsided and fails to discuss the unique technical drawbacks of using base-level coverages. While base-level coverages potentially enable discoveries like the ones discussed below, they are also much more likely to be affected by systematic technical limitations in the data (e.g. RNA-degradation), uneven contribution of 'contaminated' tissues, and differences due to amplification biases and library preparation protocols. On top of that, low-dimensional representations have the advantage of requiring fewer statistical tests and suffer less penalty due to multiple hypothesis testing. I am certain that the authors can come up with many more such examples and present their work in a more balanced context.
6. Prior knowledge in the form of gene models, while not always correct and sometimes limited, is still clearly valuable. While unbiased methods offer some advantages, biased methods (e.g. to known splice junctions) may offer higher statistical power and/or interpretability. A more nuanced discussion is needed for the relative advantages of biased/unbiased analyses.
7. The authors write that coverage is proportional to per-base expression (which is true) but hardly the only contributing factor (e.g. fragment length, 5/3 bias, PCR bias, GC bias, non-random priming). It is unclear how the method models each of those non-random effects, which can differ between samples.

8. The authors state the assumption underlying the developed method (common structure shared by the majority of samples). This assumption can and should be validated in a series of experiments: technical replicates, biological replicates, technical replicates prepared using two or more library prep protocols, a homogeneous cohort of normal samples, a heterogeneous cohort of pathological tissues (e.g. cancer samples). Though these validation experiments, it should become clear when the method is or is not applicable.

9. The concept of offnome is introduced but it is not fully clear how important it is to the method. The space devoted to offnome would be better used by better explaining the method.

10. Shape-changes can also be due to non-genetic non mutational events (should be clearly highlighted) such as differential isoform usage. This is particularly relevant in clinical samples, where differential splicing is often simply explained by varying tumor-content and admixture by non-cancer tissues. This needs to be discussed early on in the manuscript to not mislead readers/users into thinking that the method picks up only genetic events.

11. It is **exceptionally** difficult to follow the mathematical description of the method (pg 28), as almost no intuition is provided on why a particular modeling approach was taken, and few citations are provided to help the reader. For example: what are latent outlier directions, what is a spiked covariance model, and why is $g(\cdot)$ subject to normalization constraints? What are the 'd-dimensions' and what is the intuition behind X? What do the authors mean by 'dynamics' (as clearly the method is not designed for time-series)?

12. The authors introduce a new method for variance stabilization that minimizes skewness adjusted means; how critical is this method to the performance of SCISSOR? Has the method been evaluated/benchmarked in any way? It substantially adds to the difficulty of understanding this method. Perhaps the authors could first introduce a toy method to be used as baseline

13. The section on objective functions / local / global outlier detection is also very hard to follow. Perhaps the authors could introduce a toy example/method that explains the principles of their method and the idea of outlyingness in multiple dimensions?

14. Evaluation of SCISSOR is lacking as comparisons are made for a single gene, and against relatively lesser known tools (compared to the known tools cited in the introduction). Evaluations should be done in multiple cohorts, against a variety of approaches including differential exon expression, differential transcript abundance, and at a larger number of loci.

15. The authors should address the possibility that some of the "intron retention" calls may be DNA contamination. Please check the QC of these RNA-seq libraries.

16. In Figure 3b, the authors demonstrate how the baseline expression level of a gene can affect the ability to call out shape changes when there is a splice-site mutation. It is unknown how sensitive their approach is for identifying an internal deletion on a lowly-expressed gene.

17. Overall, compared to a read-based (e.g. assembly) approach, this shape-based approach could face certain challenges if the mutations have low VAF, if the samples have low purity, and/or if there is contamination from other cells (since other cells may have a different pattern of splicing).

Minor Comments:

1. Line 19: "The promise of RNA-seq remains largely unrealized" - this is a statement I would not agree with.

Reviewer #2 (Remarks to the Author):

This manuscript describes a new method, SCISSOR, to identify structural changes in RNA transcripts by shape analysis of RNA-seq read coverage profiles. This method is built upon a novel framework of detecting outlier shape changes by comparing groups of samples. The method can identify novel splicing events and transcript ends. The authors also used the method to detect "offonome", genes that are on/off within samples. Overall, the method is potentially useful in detecting abnormal RNA transcripts in large data sets, and can reveal novel events independent of known annotations. However, there are a number of issues that need to be addressed here.

The evaluation of SCISSOR is based on examination of a very small number of genes commonly known to have abnormal RNA isoforms in cancer. In comparing to other methods, only one gene was used. More systematic evaluation and comparisons are needed, such as by simulated data that can provide a genome-wide overview of performance.

The authors should evaluate sensitivity and specificity of their method, which depends on the expression level of the aberrant transcripts. In addition, the method seems to fail to identify transcript changes that only include a few nucleotides. Can the authors be more systematic in evaluating this limitation? What is the length limit and how the performance differ depending on the length of the region under consideration?

The interpretation of the results of SCISSOR seems to be quite hand-waving. For example, the claimed ATS event in FBLN5 (Figure 3) is just a speculation without further validation. In general, does SCISSOR provide a systematic way to interpret what types of events it has identified? Or is that totally up to the users and relying on manual inspection of one gene a time? If yes, this limitation greatly reduces the usability of this method.

Outlier-based methods are naturally confounded by batch effects. Can the authors elaborate how this problem is handled? Is SCISSOR only applicable to a large amount of data from the same study where batch effects are not a concern. Or can SCISSOR analyze a small number (e.g. 2-3) of RNA-seq data in the context of large reference data sets?

We thank the editor and the reviewers for what was clearly a focused effort on behalf of this manuscript. We have made major changes and added new data and analysis in response. In order to facilitate the clearest response to some concerns, we have taken the liberty of re-ordering some of the reviewers' questions and comments to group related concepts.

REVIEWER COMMENT #1:

In Choi et al., the authors present a novel method to infer changes in mRNA structures based on shape changes in an RNA-sequencing coverage profiling. They were able to identify small/large deletions, as well as mutations that may affect splicing. The methodological advance of this study is in the development of a very complex model to distinguish outlier-associated signals from random and systematic variation - including differences in sequencing depth and gene expression levels. I was very excited to try the method, and could clearly see how it may be useful. However, as presented, it is not clear (at least, not to me) as to what the method does intuitively (exposition), and how well it will function in a variety of experimental settings (validation).

We thank the reviewer for the encouragement. Both reviewers had similar comments on clarity (exposition) and we have attempted to revise the text significantly with this in mind. Questions #1, #2, #3, #5, #11, #12, #13 are specifically addressed to clarify topics in our responses. In addition, both reviewers had comments on validation and those comments were addressed in our responses to questions #14, #16, and #17. In response to these two items, the manuscript has undergone major revisions in the text and major additional analyses and data were provided.

Minor Comments:

1. Line 19: "The promise of RNA-seq remains largely unrealized" - this is a statement I would not agree with.

We agree that our statement was too strongly worded. We have changed the text as follows:

"While RNA-seq has revealed many important findings, challenges in its analysis remain."

Major comments:

1. The authors provide strong evidence that the method can pick up true signals (splice-aberrations in tumor-suppressors), but provide little evidence that the method can detect non-recurrent outlier events in passenger genes (or infrequently mutated genes) affected by mutations or structural variation.

The reviewer feels the evidence is strong for our identification of true signals in our “true positive” cases of splice site mutations. This is encouraging because identification of “truth” in genomic datasets requires some effort to establish with confidence. The use of genes for which there is a high pretest probability for any finding being a true finding (splice site mutations in tumor suppressor genes) was an approach that we found useful to prove that the method works (true positive). The reviewer did not see evidence that the method can detect non-recurrent outlier events in passenger genes or infrequently mutated genes.

The text in the section titled Genome-Wide Analysis related to figure 3 was modified to emphasize that all of figure 3 applies to the non-recurrent outlier events in which the reviewer is interested. This analysis documents that most shape changes detected by SCISSOR occur only once in a gene and are likely to be non-recurrent outlier events in passenger genes (or infrequently mutated genes) affected by mutations or structural variation.

In the section Genome-Wide analysis, the text below was added:

Importantly, this analysis emphasizes non-recurrent outlier events in both known driver (Figure 3a) and potential driver (Figure 3b) versus potential passenger genes. The distribution of the number of detected outliers was interrogated on a per gene basis in 14,399 expressed genes (after gene filtering) from the genome-wide analysis in the TCGA HNSC cohort. About 32% of the genes had at least one significant shape change, and many of them had only one or two shape changes identified. Only 2.7% of genes had alterations in > 1% of the cohort (Supplementary Note 6).

Finally, we have added a toy dataset and analysis to comment specifically on the sensitivity and false discovery rates of SCISSOR in settings of rare events. We considered different percentage levels of outlier events (5%, 2.5%, 0.5%) in 200 simulated samples and tested if there is any difference in the performance of SCISSOR. 10% and 5% indicate recurrent outlier events in 10 and 5 samples out of 200, respectively, whereas 0.5% corresponds to the case of non-recurrent event occurred only in one outlier to simulate a rare event. As shown in Extended Data Figure 4 (C) in Supplementary Note 3, we were able to observe that SCISSOR successfully identified non-recurrent events with a similar sensitivity level to the other scenarios (2.5% and 5%). More details about the simulation study will be discussed in Question #14 and Supplementary Note 3.

2. It is also unclear why the authors decided to look for TP53 outliers in HNSCC given the great recurrence of TP53 mutations in this cancer type. Indeed, most samples are mutated—which, in a way, invalidates their assumption that wild-type samples are the minority. A different cohort with a lower TP53 mutation frequency may be more appropriate.

In a prior question reviewer #1 acknowledges a point with which we agree - the value of splice-site alterations in tumor suppressors as representing a high pretest probability of “true signal.” Based on this logic, a gene like TP53 might be considered a good case study because it is frequently altered and has a high pre-test probability that alterations would be expected to be truly functional and representative of a signal that one would like to detect. The reviewer is also correct, that if the gene is too frequently altered, it might invalidate the underlying assumption that the alteration appears as an outlier in the data, but would rather assume the properties of a cluster rather than an outlier. The use of base-level data to identify clusters is one which our group has also studied and published the results [1]. Although TP53 (and CDKN2A follows a similar pattern) is frequently altered, the spectrum of individual alterations are each in themselves, rare. Individually and in small numbers, they retain the properties of outliers rather than clusters.

In other examples, however, it is clear that gene alterations, even pathologic alterations, might occur with sufficient frequency as to violate the assumption of rare outliers. As a variant becomes more common, it might be better described not as an outlier, but as a cluster and our group has proposed methods for assessment of such clusters in prior work [1]. The VIII mutation of EGFR, for example, is an example of a pathologic outlier that might assume the property of a cluster rather than an outlier.

We have added the below text to the section titled Validation on this point:

Although these genes are frequently mutated, overall no single structural alteration is recurrent or overlapping. As such, we provide evidence that the method detects non-recurrent outlier events that we extend to passenger genes and infrequently mutated genes (Supplementary Note 6).

Text changes in Methods – Questions #3,11,12,13

3. Clearly the authors have novel insights into the analysis of transcriptional profiles as well as were able to implement a statistical model and useful tool identify samples with unusual patterns. However, the method itself is not introduced in a way that would allow most readers to understand the intuition behind the approach. While the statistics and equations are reasonably well explained, the manuscript is riddled with difficult concepts, e.g.: “A latent variable is used to model an underlying abnormal trajectory, i.e. an outlier direction in a high-dimensional space, that is interrogated for outliers.”

In comments #3,#11,#12, and #13, as well as in the overall comments from reviewer #1, the reviewer feels that both in general and in specific, the exposition of the methods requires further clarification. To address this, we have significantly altered the text. We have focused on exposition for each step of the method and moved most of the technical jargon and notation to the Methods. In cases where we have retained terms of art (or jargon), we have endeavored to define these each explicitly including those from the phrase cited by the reviewer: “A latent variable is used to model an underlying abnormal trajectory, i.e. an outlier direction in a high-dimensional space, that is interrogated for outliers.” In Methods, we made significant changes in the section titled Modeling base-level read counts. Also, to clarify the exposition in main text, we added a new section below (Pre-processing procedure) and edited the section titled The proposed model:

Pre-processing procedure

Before the main shape change detection procedure, normalization and variance stabilization are performed to generate more symmetrical distributions of the SCISSOR test statistic and to facilitate interpretation of statistical testing across genes. Because SCISSOR is unlike most gene expression methods in considering base level gene expression as opposed to summary gene expression or expression of exons or splice junctions, modifications of existing normalization and variance stabilization methods were required. We perform additional pre-processing that can be thought of as determining the residual per-base expression for each sample after subtraction of a model representing the shape of that gene across all samples (figure1b). We refer to a potential abnormal residual structure as latent gene expression because it is not directly observed but inferred after subtraction of the model from the normalized data.

In main text, we also have reproduced the edited section below for the reviewer’s convenience.

The proposed model

The curve after the normalization (Fig. 1b) can be viewed as a single point in a high dimensional space by considering each base position as a dimension and the height of latent gene expression as a value at that position. By considering each curve in the normalized data as a random vector in a high-dimensional space, we fit multiple unknown mixture distributions, recasting the problem into a high-dimensional latent variables framework (Methods). A latent variable is used to model an underlying abnormal gene expression trajectory, i.e. an outlying direction in a high-dimensional space, that is interrogated for outliers. An outlier case with shape changes then can be a data point that is strongly involved in one or multiple abnormal trajectories, which enables modeling complex structural variation.

11. It is *exceptionally*** difficult to follow the mathematical description of the method (pg 28), as almost no intuition is provided on why a particular modeling approach was taken, and few citations are provided to help the reader. For example: what are latent outlier directions, what is a spiked covariance model, and why is $g(\cdot)$ subject to normalization constraints? What are the ‘d-dimensions’ and what is the intuition behind X ? What do the authors mean by ‘dynamics’ (as clearly the method is not designed for time-series)?**

This comment seems to be related to comment #3 above. We have re-written an exposition designed to engage readers whose training does not include the full range of techniques required for the method. In the methods, we have gone through line by line to re-write such that all the terms the reviewer mentions in this comment or other comments are specifically defined in the text.

In Methods, we made significant changes in the section titled Modeling base-level read counts and added more references.

12. The authors introduce a new method for variance stabilization that minimizes skewness adjusted means; how critical is this method to the performance of SCISSOR? Has the method been evaluated/benchmarked in any way? It substantially adds to the difficulty of understanding this method. Perhaps the authors could first introduce a toy method to be used as baseline.

We agree that the manner in which we present variance stabilization was incomplete and confusing. Variance stabilization is not new, and neither is addition of pseudocount. Selection of pseudocount offset is often empiric such as in the reference provided[2]. Rather than selection of a single empiric offset parameter for the entire dataset (such as the addition of 1 to all counts) we have optimized the offset to minimize skewness using an established statistic for normality, the anderson-darling statistic. Symmetric variance is a property which is inherently desirable to the family of test statistics used such that a formal proof would not generally be required. Additionally, we select the offset on a gene by gene basis since the gene is the unit of analysis in this study.

We have re-written the text and added appropriate references.

In Supplementary Note 1 - Normalization section:

Our group and others have previously shown that when the read counts are low or highly variable, estimates of gene expression have high variance compared to genes with higher expression. To avoid spurious detection of differences, investigators must either remove such genes by filtering or adopt variance stabilization approaches such as pseudocounts[3-5]. The use of pseudocounts are preferred since solutions have been proposed to model stabilized variance, such as assessment of symmetry around the median. Additionally, variance stabilization obviates the need to discard genes through filtering[2]. Whereas most studies parameterize pseudocounts at the experimental level by selection of a single offset value, our purpose of characterizing gene by gene shape changes requires variance stabilization at the gene level. For the purposes of variance stabilization we optimize the A-D statistic as a function of pseudocount offset.

13. The section on objective functions / local / global outlier detection is also very hard to follow. Perhaps the authors could introduce a toy example/method that explains the principles of their method and the idea of outlyingness in multiple dimensions?

The manner in which we present the objective function was not clear. Importantly, we had already provided a toy example (figure 1) which was not clearly referenced in the text relative to the methods, and we hope that by calling this to the reader's attention more clearly the section will be easier to understand.

In addition, we have added a new toy example to the supplemental notes. The section titled Modified projection outlyingness in Methods was also re-written to reference Figure 1 to illustrate the toy example. In main text, we also have reproduced the edited section below for the reviewer's convenience.

Shape change detection

SCISSOR extracts a latent gene expression associated with abnormal sequencing coverage and quantifies the level of abnormality in a robust way for determining the cases with shape changes (Fig. 1c,d). As the type of structure of interest here is outlying/abnormal, we search for the best one-dimensional projection in which each sample achieves the maximum deviation from all other samples, as a result, producing a direction we termed the "most outlying direction" (MOD). Recognizing that RNA-seq data may retain the quality of skewness despite normalization, we propose a modified projection outlyingness approach which is complementary to a robust measure of how outlying a sample is in the most extreme one-dimensional direction [6, 7] (Methods, Supplementary Note 5). This modified approach helps to avoid spurious outliers due to strong skewness of distributions as well as enables approximate comparisons of outlyingness across genes and samples. At each gene under consideration, the resulting statistic is an outlyingness score for each sample with larger values indicating more severe deviation from other samples in the dataset (Fig. 1c). The observed distribution of the outlyingness scores can be used for identifying and ranking outliers and modeling statistical significance. For each outlier, SCISSOR produces the MOD as a single best trajectory that describes abnormalities of the corresponding outlier, which can be used to recover the latent space of underlying outlier directions (Methods, Supplementary Fig. 1).

The current method increases the dimensionality of the data by considering expression at a per-base level along the entire length of the gene under consideration. Although this allows high-resolution views of RNA variation, it also entails a higher level of noise due to sampling variation. Accordingly, our approach must

address the signal-to-noise ratios generated by accumulated noise in such a high dimensional framework. Specifically, we attempt to detect with similar success, both focal and broad structural alterations in genes expressed either at very high/very low levels, as well as for very short/very long regions, challenges which we feel are not well captured by current methods.

To address these challenges, SCISSOR implements a two-step procedure, global and local, taking advantage of low-dimensional transformations and sparsity (Methods). We define a global shape change (GSC) as one that appears in a wide range within a gene, including altered exons or introns, intragenic deletions, and fusions. The directions supporting disjoint exonic or intronic regions are considered as a set of orthogonal bases of a low rank space that will be searched by the modified projection outlyingness method. As a result, the MOD of an outlier detected by the global shape change detection is established by a linear combination of the bases (or a basis itself), which enables the interpretation of the abnormality and the automatic characterization of its type (Methods, Supplementary Fig. 1a-c, Supplementary Table 1). By contrast, the local shape change (LSC) detection procedure only considers changes in closely related base positions. In this second step designed to detect more focal sequence changes often missing from the low rank space, we interrogate the remaining residuals. We optimize the residual outlyingness with respect to sparse directions supporting important regions within a given gene. This helps detect local genomic variation such as abnormal gains or losses at narrow regions (Supplementary Fig. 1d-e).

4. It is unclear how big the cohort-size has to be for SCISSOR to work. It also not explained whether it can be used in a case-control setting and with a limited number of replicates.

The reviewer reasonably requests guidance on sample sizes needed to expect performance from SCISSOR. The following text was added to the Methods (at the end of the section – Modified projection outlyingness).

The concept of an outlier is with respect to an expected distribution. As such any cohort to which SCISSOR is applied should be sufficiently large to assess the underlying distributions and any outlier of interest should be sufficiently large to be detected in a cohort of that size. Precise experiments to measure effect size and distributions are the subject of future work, but empirically a cohort of 30 samples would be the smallest we have considered successfully to date.

The current version of SCISSOR does not provide a direct comparison between case/control groups. However, it is possible to compare the results from the two groups by using statistical tests. For example, exon 14 skipping in *MET* was observed in 4% (10/230) of TCGA LUAD tumor samples but none in wild type (0/199) [8]. We applied SCISSOR to this cohort and identified these 10 cases with the *MET* exon 14 skipping with p-value=0.002 from the Fisher's exact test, indicating that there is a significant difference between tumor/normal groups. In this manner, we can use SCISSOR to see if specific outliers are dependent on case groups.

Comparison / Validation – Questions #14,16

14. Evaluation of SCISSOR is lacking as comparisons are made for a single gene, and against relatively lesser known tools (compared to the known tools cited in the introduction). Evaluations should be done in multiple cohorts, against a variety of approaches including differential exon expression, differential transcript abundance, and at a larger number of loci.

The reviewer feels that the comparator methods were not optimally selected. We suspect that we failed to describe the goals and the challenges of selecting comparator methods. We have altered the text to clarify the challenges and the goals.

We added the below text to Supplementary Note 3:

Although SCISSOR is an unsupervised outlier detection method designed to interrogate expressed gene variants across the entire gene, biologic interpretation of gene outliers is often in terms of alternative splicing or exon expression at a single locus or region. Detection of alternative splicing and exon expression is the outcome of many supervised RNA-seq algorithms, and as such, we considered a comparison of the results of SCISSOR versus more common approaches to detection of exon skipping and alternative splicing. A direct comparison was not obvious, given differences between unsupervised methods and supervised methods, as well algorithmic modifications and normalization required to articulate an objective comparison. However, it was possible to adapt the statistical framework underlying the two most common approaches to alternative splicing and gene expression using junction reads-based analysis (JRBA) and exon level expression-based analysis (ELBA). JRBA and ELBA are explicitly involved as the framework for most if not all relevant comparator methods including the percent-spliced-in and DEXSeq [9-13].

The reviewer also asks that we perform this evaluation in multiple cohorts, at differential transcript abundance, and at a larger number of loci.

This point was also raised by reviewer #2 in question #1. We have added a significant amount of new data and analysis to address this shortcoming to Supplementary Note 3.

16. In Figure 3b, the authors demonstrate how the baseline expression level of a gene can affect the ability to call out shape changes when there is a splice-site mutation. It is unknown how sensitive their approach is for identifying an internal deletion on a lowly-expressed gene.

We appreciate the reviewer's comment, which was partially the intention of figure 3b. It is true that at lower levels of gene expression, shape changes of all types are observed less frequently. It is important to note that there are at least two interpretations of this finding, only one of which is lower sensitivity. The other interpretation is that some base substitutions likely did not result in large shape changes and might therefore be considered passenger mutations.

Based on this comment and others such as reviewer #1, question #14 and reviewer #2, question #1, we added a new toy dataset and analysis in which we introduce specific events of run on exons and exon skipping to simulate the functional consequences of splice site alterations. The simulation varies the percentage of alleles impacted by the shape changes as a total of overall gene expression and varies the overall expression of the gene to assess sensitivity across a range of conditions. See Supplementary Note 3 – Simulation study.

***Technical biases and other challenges* - Questions #5, 7, 10, 17, 6, 8, 15**

5. The introduction is lopsided and fails to discuss the unique technical drawbacks of using base-level coverages. While base-level coverages potentially enable discoveries like the ones discussed below, they are also much more likely to be affected by systematic technical limitations in the data (e.g. RNA-degradation), uneven contribution of ‘contaminated’ tissues, and differences due to amplification biases and library preparation protocols. On top of that, low-dimensional representations have the advantage of requiring fewer statistical tests and suffer less penalty due to multiple hypothesis testing. I am certain that the authors can come up with many more such examples and present their work in a more balanced context.

The reviewer finds the introduction unbalanced related to failing to discuss drawbacks of base level coverage. We agree with the reviewer about the importance of discussing the relevant shortcomings of a base level approach, and we feel that many of these are demonstrated explicitly in the manuscript analyses. With the indulgence of the reviewer, and in the interest of brevity and clarity, we discuss shortcomings primarily in the discussion where we reference examples shown by our own work and include others beyond the scope of this manuscript.

The following text was added to Discussion:

Although we have shown that consideration of base-level coverage offers the potential to detect known and novel classes of variants, there remain a number of challenges. Systematic coverage bias may occur within a single sample or batch of a cohort due to RNA degradation or fragment length or other unknown factors. We have shown the ability to detect samples with degraded RNA across genes in specific samples. After removing such samples, we then attempted to detect residual RNA degradation in calls of outlier genes as assessed by clustering outlier genes and samples. No evidence of residual RNA degradation was observed, although we recognized that our inability to detect it does not exclude the possibility of its existence. Likewise, although TCGA samples such as we have used have high quality RNA, it is likely that factors such as variable fragment length would introduce artifactual signals detectable by SCISSOR. The impact of GC content on nucleic acid sequencing is well known to reduce template amplification, and as such introduce artifactual shape changes. When such changes are systematic across all samples, they will be removed by normalization, but if experimental conditions such as temperature are varied, we might observe sample specific effects as well. To the extent that these effects are measurable by SCISSOR, they will complicate the biologic interpretation of any results.

7. The authors write that coverage is proportional to per-base expression (which is true) but hardly the only contributing factor (e.g. fragment length, 5/3 bias, PCR bias, GC bias, non-random priming). It is unclear how the method models each of those non-random effects, which can differ between samples.

The reviewer is referencing well-known challenges in RNA analysis, a complete investigation of which is challenging to present in this first manuscript. Having said that, many related concerns were evaluated in the initial work on SCISSOR and are presented in the methods and

analysis, although perhaps with insufficient clarity based on the reviewer's comment. We have modified the discussion to discuss additional contributing factors governing base coverage as suggested.

The following text was added to Discussion:

In the current work we describe a novel method for detecting outlier RNA events across samples. Changes may be due to physiologic events, such as alternate splicing, pathologic events, such as mutation, or experimental factors such as RNA degradation or contaminating stromal cells. To the extent that outlier shapes are detected by SCISSOR as represented by the underlying data, we consider this a success of the method, even when those changes might relate to experimental factors such as RNA degradation (clearly seen in the data), contaminating stroma cells (likely observed in the current dataset and the source of future reports), GC content (potentially seen in the current report as described), or RNA fragment length (not obviously detected in the current report but potentially a concern). Users of SCISSOR should consider experimental factors as well as biological when interpreting the results of SCISSOR. In this work, we observed that RNA degradation, if not addressed, will result in degraded samples producing large numbers of outlier genes. We addressed this by empirically removing samples with evidence of degradation. We observed some weaker evidence coordinated RNA shapes shared by samples across many genes in association with certain types of internal CpG islands. Whether or not this is a GC experimental artifact or a biologic process is unclear from the current data.

10. Shape-changes can also be due to non-genetic non mutational events (should be clearly highlighted) such as differential isoform usage. This is particularly relevant in clinical samples, where differential splicing is often simply explained by varying tumor-content and admixture by non-cancer tissues. This needs to be discussed early on in the manuscript to not mislead readers/users into thinking that the method picks up only genetic events.

We agree wholeheartedly with this point although pursuit of specific and definitive examples can be challenging. For the reviewer, rather than differential isoforms from contaminating cells, we are more convinced of the presence of admixtures of non-cancer tissue when we see the presence of rare transcripts associated with the contaminating cells rather than alternate splice forms. We do quite definitively see alternative splices that are non-mutations (including recurrent events in multiple patients). However, these events are generally associated with a germline genetic event such as splice-site alteration reported in population databases or in the case of the gene LSS (figure 3e), a germline CNV (For more details, please see the section titled Genome-wide analysis). At the recommendation of the reviewer, the introduction was modified as follows:

RNA changes may be somatic, as the result of driver or passenger mutations, germline variants, or non-genetic events resulting from epigenetic regulation of alternate isoforms.

17. Overall, compared to a read-based (e.g. assembly) approach, this shape-based approach could face certain challenges if the mutations have low VAF, if the samples

have low purity, and/or if there is contamination from other cells (since other cells may have a different pattern of splicing).

The reviewer correctly considered the impact of low purity on the sensitivity of SCISSOR. We have added new toy examples and analysis to address the impact of decreased allele fraction on SCISSOR. In fact, SCISSOR compares favorably to competing methods as a function of low allele fraction and low gene expression. We added extensive experiments in this regard, demonstrating that, overall, SCISSOR performs well compared to competing methods, although we acknowledge 2 specific weaknesses. If alternative splicing or splice-site mutations result in changes of only a few bases (for example <20) in the spliced position, the RNA shape change is difficult for SCISSOR to detect. Changes in the parameterization of SCISSOR's local outlier detection might be able to improve this weakness (Supplementary Note 7). Secondly, for genes expressed at very high levels, we require a larger variant allele fraction to detect the shape change related to the impact of log transformation. Again, changes in the parameterization are able to increase the power of SCISSOR in this scenario, although with negative impacts on specificity in other cases.

To address this concern we have added new data as a toy example, text, analyses, and figures to describe the performance of SCISSOR as a function of variant allele fraction for different classes of shape changes and at varying levels of gene expression. See Simulation study in Supplementary Note 3.

6. Prior knowledge in the form of gene models, while not always correct and sometimes limited, is still clearly valuable. While unbiased methods offer some advantages, biased methods (e.g. to known splice junctions) may offer higher statistical power and/or interpretability. A more nuanced discussion is needed for the relative advantages of biased/unbiased analyses.

We thank the reviewer for this comment which allows us to offer a more nuanced view of SCISSOR than was presented in our first submission. In theory, SCISSOR does not require gene models and can be applied directly to the genome. However, the interpretation of gene expression without regard to existing gene models is challenging such that we have elected to implement SCISSOR within the context of gene models. Additionally, the interpretation of any altered shape can be challenging without the context of splicing. While splicing is not a component of the statistical test, it is incorporated in the interpretation of the biology of shape changes. We now appreciate that we are using the term unbiased in a somewhat different manner than is usually intended for RNA transcript isoform analysis. In the context of SCISSOR unbiased refers to the fact that although we are only looking in the space of existing gene models, RNA coverage is not constrained to prespecified exon start or stop locations and may continue into introns. The following text was added to the Methods (to the section – Inclusion of intronic part):

Although SCISSOR does not require a gene model for most of its underlying assumptions, gene models greatly facilitate the biologic interpretation and visualization of results. The SCISSOR procedure, including normalization and outlier detection can be applied directly to RNA aligned to the genome in a truly unbiased manner. However, such a procedure can produce results that are difficult to interpret as RNA is most easily described in the context of named genes, exons, introns, and splicing. As such, the

SCISSOR procedure is applied in the context of named gene models to assist in visualization and interpretation of the data. SCISSOR is unbiased in that space because it is not constrained by existing exon start or stop positions, and incorporates intron space to allow detection of events outside exons such as run on transcriptions. Additionally, although SCISSOR does not include splicing information in the statistical procedure, it does incorporate splices in the interpretation of shape changes once they are identified as described in the section titled “Most outlying direction” in Methods.

8. The authors state the assumption underlying the developed method (common structure shared by the majority of samples). This assumption can and should be validated in a series of experiments: technical replicates, biological replicates, technical replicates prepared using two or more library prep protocols, a homogeneous cohort of normal samples, a heterogeneous cohort of pathological tissues (e.g. cancer samples). Though these validation experiments, it should become clear when the method is or is not applicable.

Although we have reported the use of shared and discrepant RNA pileup figures on a number of occasions we have never formally characterized or defined the concept of “common structure” and thank the reviewer for this opportunity. We added references to other manuscripts where the concept of “common structure” was reported to demonstrate shared and discrepant RNA variants [1, 14-16]. To formally evaluate the extent to which transcripts assessed by RNA have common structure we have generated the analyses requested through technical and biological replicates in normal and pathologic tissues. We have added text, methods, figures, analysis, and interpretation to the Supplementary Note 8.

15. The authors should address the possibility that some of the "intron retention" calls may be DNA contamination. Please check the QC of these RNA-seq libraries.

The reviewer correctly identifies DNA contamination as a potential source of intronic sequences. The following text was added to the discussion.

For those samples with evidence of retained intron we considered the possibility of DNA contamination as an explanation. DNA contamination as assessed by the 260/280 ratio was consistent with pure RNA. Additionally, we would expect that contaminating DNA would be evidence as a general phenomenon across many introns at lower or consistent levels rather than as we see it in highly selected introns at high coverage in a very limited number of introns.

9. The concept of offnome is introduced but it is not fully clear how important it is to the method. The space devoted to offnome would be better used by better explaining the method.

We agree with the reviewer. The offnome section is removed.

REVIEWER COMMENT #2:

This manuscript describes a new method, SCISSOR, to identify structural changes in RNA transcripts by shape analysis of RNA-seq read coverage profiles. This method is built upon a novel framework of detecting outlier shape changes by comparing groups of samples. The method can identify novel splicing events and transcript ends. The authors also used the method to detect “offonome”, genes that are on/off within samples. Overall, the method is potentially useful in detecting abnormal RNA transcripts in large data sets, and can reveal novel events independent of known annotations. However, there are a number of issues that need to be addressed here.

We thank the reviewer for the encouragement. We have added major new analyses to explore the performance of the methods by comparing with other methods and evaluating sensitivity and false discovery rate as the reviewer requests. In addition, we have made provided new analyses to improve interpretability of the types of the identified shape outliers by incorporating the splice junction reads data. Last, we have added a comment on the batch effects that might confound shape outliers to Discussion.

In order to address the range of clarifications of reviewer #1, in consideration of the manuscript length, and at the suggestion of reviewer #1 we have removed the section on the “offonome.” We plan to report that in a separate manuscript.

#1. The evaluation of SCISSOR is based on examination of a very small number of genes commonly known to have abnormal RNA isoforms in cancer. In comparing to other methods, only one gene was used. More systematic evaluation and comparisons are needed, such as by simulated data that can provide a genome-wide overview of performance.

This comment is very similar to concerns raised by Reviewer #1 in question #14, #16, and #17. Please see those replies for details. Having said this, we have added significant amount of data, text, methods, analysis, and figures to address the systematic evaluation in comparison to other methods. This is primarily located in Supplementary Note 3.

#2. The authors should evaluate sensitivity and specificity of their method, which depends on the expression level of the aberrant transcripts. In addition, the method seems to fail to identify transcript changes that only include a few nucleotides. Can the authors be more systematic in evaluating this limitation? What is the length limit and how the performance differ depending on the length of the region under consideration?

The reviewer correctly observes a specific shortcoming of SCISSOR as reported in the first draft which was presented only briefly. We have generated new text, data, simulations, figures, and analysis to provide additional insights into this weakness. Please see Supplementary Note 3 in the revised manuscript.

#3. The interpretation of the results of SCISSOR seems to be quite hand-waving. For example, the claimed ATS event in FBLN5 (Figure 3) is just a speculation without further validation. In general, does SCISSOR provide a systematic way to interpret what types of events it has identified? Or is that totally up to the users and relying on manual inspection of one gene a time? If yes, this limitation greatly reduces the usability of this method.

The reviewer is correct that in the previous draft we emphasized the search for outliers over their interpretation. We appreciate the reviewers request to address this shortcoming and to that end we have added significant new text, methods, analysis, interpretation, and a table. Please see these changes in the section titled the “Shape change detection” in the main draft, the sections titled “Global shape change detection” and “Most outlyingness direction” in Methods.

#4. Outlier-based methods are naturally confounded by batch effects. Can the authors elaborate how this problem is handled? Is SCISSOR only applicable to a large amount of data from the same study where batch effects are not a concern. Or can SCISSOR analyze a small number (e.g. 2-3) of RNA-seq data in the context of large reference data sets?

The reviewer asks about a common concern for computational biology, the potential role of batch effects. Additionally, the reviewer includes a question that we interpret to be similar to question #5 of Reviewer #1 on the topic of sample size. Since one solution to the problem of batch effects is to perform stratified analysis by batch, the ability of SCISSOR to analyze smaller batches is key.

As with many techniques in computational biology, the possibility that experimental or other batch effects might associate with results is a concern. The unit of analysis in the current study is the gene as a function of samples. We specifically considered those types of batch effects which would lead us to conclude that a sample was an outlier when in fact it was due to a batch effect. We anticipated that batch effects would take the following form: outlier samples would appear outlier in a common set of genes defined by the batch effect. By looking across genes and within samples, such coordination might be interpreted as shared biology but might also be due to artifactual experimental or sample effects. We clearly identified one such effect associated with RNA degradation, and opted to treat degraded samples as a batch by excluding them from further analysis as described in the methods. RNA degradation did not appear to be an experimental batch effect associated with assay production date, but rather with degradation inherent to the sample, likely associated with the time of collection. We identified a second set of shared outliers across samples as identified in figure 4. In this case, the outlier events appeared to associate with changes in gene structure related to CpG islands, suggesting that an underlying biologic process might be at play. Alternatively, we recognize that GC content can be associated with differential PCR efficiency, and an experimental batch effect would be an alternative explanation. Consideration of assay production date did not clearly explain this effect. It is likely that other technical and biologic factors might be at work in these data or other datasets to account for outlier genes detected by SCISSOR, and that the user should consider a wide range of causes for those shape changes reported by the algorithm. Importantly in the TCGA dataset we have some assurances that experimental batch effects have been rigorously investigated and largely excluded (See supplement S10 in [15]).

We refer the Reviewer to the response to question #5 of Reviewer #1 for additional relevant comments.

1. Kimes, P.K., et al., *SigFuge: single gene clustering of RNA-seq reveals differential isoform usage among cancer samples*. Nucleic Acids Res, 2014. **42**(14): p. e113.
2. Zhu, A., J.G. Ibrahim, and M.I. Love, *Heavy-tailed prior distributions for sequence count data: removing the noise and preserving large differences*. Bioinformatics (Oxford, England), 2019. **35**(12): p. 2084-2092.
3. Law, C.W., et al., *voom: Precision weights unlock linear model analysis tools for RNA-seq read counts*. Genome biology, 2014. **15**(2): p. R29-R29.
4. Ritchie, M.E., et al., *limma powers differential expression analyses for RNA-sequencing and microarray studies*. Nucleic acids research, 2015. **43**(7): p. e47-e47.
5. Robinson, M.D., D.J. McCarthy, and G.K. Smyth, *edgeR: a Bioconductor package for differential expression analysis of digital gene expression data*. Bioinformatics (Oxford, England), 2010. **26**(1): p. 139-140.
6. Dang, X. and R. Serfling, *Nonparametric depth-based multivariate outlier identifiers, and masking robustness properties*. Journal of Statistical Planning and Inference, 2010. **140**(1): p. 198-213.
7. Donoho, D.L. and M. Gasko, *Breakdown properties of location estimates based on halfspace depth and projected outlyingness*. The Annals of Statistics, 1992. **20**(4): p. 1803-1827.
8. Cancer Genome Atlas Research, N., *Comprehensive molecular profiling of lung adenocarcinoma*. Nature, 2014. **511**(7511): p. 543-550.
9. Seiler, M., et al., *Somatic mutational landscape of splicing factor genes and their functional consequences across 33 cancer types*. Cell reports, 2018. **23**(1): p. 282-296. e4.
10. Song, Y., et al., *Single-cell alternative splicing analysis with expedition reveals splicing dynamics during neuron differentiation*. Molecular cell, 2017. **67**(1): p. 148-161. e5.
11. Anders, S., A. Reyes, and W. Huber, *Detecting differential usage of exons from RNA-seq data*. Genome research, 2012. **22**(10): p. 2008-2017.
12. Climente-González, H., et al., *The functional impact of alternative splicing in cancer*. Cell reports, 2017. **20**(9): p. 2215-2226.
13. Katz, Y., et al., *Analysis and design of RNA sequencing experiments for identifying isoform regulation*. Nature methods, 2010. **7**(12): p. 1009.
14. *Comprehensive genomic characterization of squamous cell lung cancers*. Nature, 2012. **489**(7417): p. 519-25.
15. Network, C.G.A., *Comprehensive genomic characterization of head and neck squamous cell carcinomas*. Nature, 2015. **517**(7536): p. 576.
16. *Comprehensive molecular profiling of lung adenocarcinoma*. Nature, 2014. **511**(7511): p. 543-50.

REVIEWERS' COMMENTS

Reviewer #1 (Remarks to the Author):

The reviewers have address all previous concerns, no additional comments.

Reviewer #2 (Remarks to the Author):

None